# BioArc: Discovering Optimal Neural Architectures for Biological Foundation Models

## Abstract

Foundation models have revolutionized various fields such as natural language processing (NLP) and computer vision (CV). While efforts have been made to transfer the success of the foundation models in general AI domains to biology, existing works focus on directly adopting the existing foundation model architectures from general machine learning domains without a systematic design considering the unique physicochemical and structural properties of each biological data modality. This leads to suboptimal performance, as these repurposed architectures struggle to capture the long-range dependencies, sparse information, and complex underlying "grammars" inherent to biological data. To address this gap, we introduce BioArc, a novel framework designed to move beyond intuition-driven architecture design towards principled, automated architecture discovery for biological foundation models. Leveraging Neural Architecture Search (NAS), BioArc systematically explores a vast architecture design space, evaluating architectures across multiple biological modalities while rigorously analyzing the interplay between architecture, tokenization, and training strategies. This large-scale analysis identifies novel, high-performance architectures, allowing us to distill a set of empirical design principles to guide future model development. Furthermore, to make the best of this set of discovered principled architectures, we propose and compare several architecture prediction methods that effectively and efficiently predict optimal architectures for new biological tasks. Overall, our work provides a foundational resource and a principled methodology to guide the creation of the next generation of task-specific and foundation models for biology.

## 1 Introduction

The advent of foundation models, large-scale neural networks pretrained on vast amounts of data, has catalyzed a profound revolution in artificial intelligence (AI). In general domains such as natural language processing (NLP) (Devlin et al., 2019; OpenAI, 2023; Touvron et al., 2023) and computer vision (CV) (Dosovitskiy, 2020; Liu et al., 2024), foundation models built on top of architectures such as Transformers (Vaswani et al., 2017) and Diffusion Models (Ho et al., 2020) have demonstrated unprecedented capabilities, fundamentally reshaping research and applications. This success has naturally spurred a wave of interest in applying this paradigm to biology, where large-scale datasets are becoming increasingly available due to years of effort from biological scientists in collecting and organizing the data into large-scale databases. The promise is to leverage the existing large data to create powerful biological foundation models (Xiao et al., 2025) that can learn the generative grammar of genetic or protein sequences, accelerating discovery in drug development, synthetic biology, and personalized medicine.

However, directly applying architectures from general AI domains presents numerous limitations to the biological domain. Most current biological foundation models (Zhou et al., 2024; Rives et al., 2021; Xiao et al., 2025) are built upon the Transformer architecture (Vaswani et al., 2017). However, Transformers were originally designed for human language and not for the complex "grammar" unique to biological data, making them potentially suboptimal for biological applications.

For example, in contrast to the grammatical rules of text, biological sequences are governed by a set of complex, hierarchical rules dictated by physicochemical laws (Dill & MacCallum, 2012). To be more specific, they contain non-local dependencies (e.g., structural motifs) (Wang et al., 2017),

have varying lengths, and their information is encoded in a discrete alphabet with a specific biological "grammar" (Greener et al., 2022). Simply re-purposing existing architectures risks suboptimal performance and a failure to capture the complex biological principles encoded within these sequences. This leads to a key research question: *What constitutes an optimal neural architecture for a biological foundation model?*

Answering this question is non-trivial as there exist several technical challenges. Most importantly, architectural innovation in this domain is hampered by a lack of a comprehensive set of guiding principles for each data modality. While researchers may possess some partial knowledge, such as the physical laws governing protein design (Marcos et al., 2017), our fundamental scientific understanding of complex biological systems remains limited and is still a subject of ongoing research. This presents a stark contrast to a field like NLP, where language is a human-generated system with rules and structures we inherently understand. Biological information, on the other hand, is designed by nature; its underlying principles are yet to be fully discovered and understood by humans, a journey that is still a long way from its destination. Consequently, unlike the NLP field, there is no universally acknowledged architecture that consistently excels across the board. This forces a reliance on manual, intuition-driven design, a process that is inefficient and has yet to yield a dominant architectural blueprint for biological foundation models (Sapoval et al., 2022; Eisenstein, 2024; Vishniakov et al., 2025). In addition, and adding to the complexity, is the deep entanglement of architecture, data preprocessing, and training strategy. The efficacy of a given architecture can be dramatically altered by choices in tokenization and optimization schemes, meaning that architectural choices cannot be meaningfully designed, developed, and evaluated standalone.

To address these challenges, we introduce **BIOARC**, a framework advancing from intuition-driven design to principled, automated discovery. Leveraging Neural Architecture Search (NAS), BIOARC systematically evaluates how blocks like CNNs and Transformers compose to model biological data. Recognizing that performance depends on modality, tokenization, and training, we design BIOARC as a unified testbed to analyze this interplay. Thus, BIOARC not only identifies high-performance architectures but also reveals design patterns for future foundation models. Crucially, we incorporate predictors to efficiently select optimal architectures for new tasks. Our contributions are threefold: First, BIOARC achieves state-of-the-art performance with models $25\times$ smaller. Second, we systematically evaluate training and tokenization strategies across modalities to guide future design. Finally, we benchmark methods for predicting optimal biological architectures.

## 2 RELATED WORK

**Foundation Models for Biological Data**    Inspired by NLP and CV, biological foundation models learn patterns from vast unlabeled sequences via self-supervised pretraining. In proteomics, transformer-based models like AlphaFold (Jumper et al., 2021) revolutionized structure prediction. Scaling to the whole-genome level, Evo-1 and Evo-2 (Nguyen et al., 2024; Brixi et al., 2025) enables generative design. For sequence representation, models such as PathoLM (Dip et al., 2024), Nucleotide Transformer (Dalla-Torre et al., 2023), Gena-LM (Fishman et al., 2023), and VQDNA (Li et al., 2024) target DNA; ProtBert (Elnaggar et al., 2021) and ESM-1 and ESM-2 (Rives et al., 2021; Lin et al., 2023) focus on proteins; while RNAErnie (Wang et al., 2024) and RNABERT (Akiyama & Sakakibara, 2022) address RNA. Expanding modalities, Geneformer (Theodoris et al., 2023) and CellFM (Zeng et al., 2025) targets single-cell gene expression value sequence.

**Hybrid Architecture**    Beyond pure sequence models, hybrid architectures combine distinct neural mechanisms to leverage complementary strengths. In protein science, integrating GNNs with attention captures both sequential context and geometric constraints (Jumper et al., 2021). Similarly, for nucleic acids, merging CNNs with Transformers enables efficient processing of high-resolution sequences; CNNs encode local elements and reduce length, allowing Transformers to model distal interactions such as enhancer-promoter contacts (Avsec et al., 2021; Yu et al., 2024).

**Neural Architecture Search**    Neural Architecture Search (NAS) automates architecture design. While early RL (Zoph & Le, 2017) and evolutionary (Real et al., 2017) methods were computationally expensive, efficient one-shot techniques (Liu et al., 2019) have since enabled landmarks like EfficientNet (Tan & Le, 2020) on ImageNet (Deng et al., 2009). However, biological applications remain limited. Prior works focus mostly on single architectures (Zhang et al., 2021b;a) or spe-

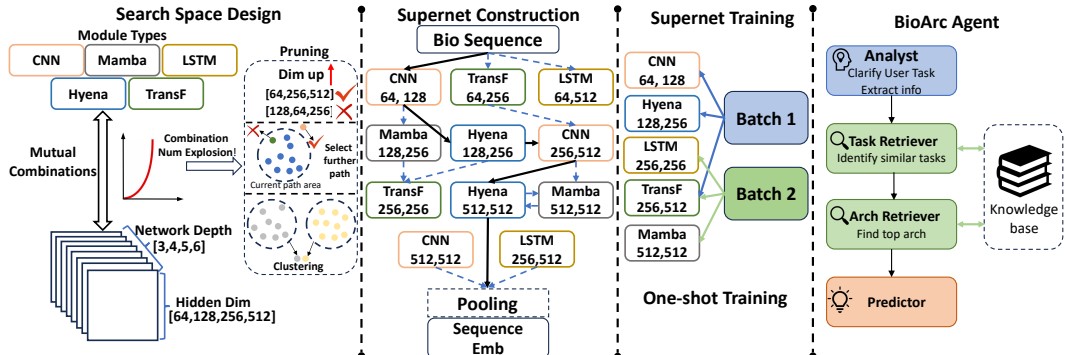

Figure 1: This figure provides an overview of the four core stages of the BioArc framework. (1) Search Space Design: Defines diverse module types, network depths, and hidden dimensions, using pruning and clustering strategies to manage the combinatorial explosion. (2) Supernet Construction: Encodes the vast search space into a single, weight-sharing Supernet, with each path representing a candidate architecture. (3) Supernet Training: Adopts a one-shot methodology, sampling and optimizing different paths across batches for efficiency. (4) BioArc Agent: An intelligent agent that analyzes user tasks, leverages a knowledge base to retrieve similar tasks and top architectures, and predict top architectures from the supernet.

cific sequence tasks (Sivangi et al., 2022). Even GenomeNet-Architect (Gündüz et al., 2024), which combines CNNs and RNNs, is constrained by a restricted search space of module types.

**Architecture Prediction** To mitigate the high computational cost of NAS, architecture predictors serve as efficient surrogates, estimating performance directly from structural properties (Ying et al., 2019). Trained on pre-evaluated architecture-performance pairs, these models range from simple regressors to Graph Neural Networks capable of encoding complex topologies (Łukasz Dudziak et al., 2021). However, their generalization is inherently limited by the scope of training tasks (Duan et al., 2021), restricting transferability to novel domains without significant retraining.

## 3 BIOARC FRAMEWORK

This chapter introduces BIOARC, a framework for systematically exploring biological neural architectures. To analyze design-performance relationships, BIOARC is structured around following dimensions: architecture search space, tokenization methods, training strategies, and downstream tasks. While demonstrated on DNA and protein, the framework generalizes to other modalities such as RNA and single-cell data.

### 3.1 BENCHMARK TASKS AND DATASETS

**DNA** For pretraining, we utilize the full human reference genome **GRCh38**. For downstream evaluation, we employ the human subset of the **GUE benchmark** (Zhou et al., 2024), comprising four tasks across 12 distinct datasets. Detailed descriptions are provided in Appendix A.1.1.

**Protein** For pretraining, we use a sampled subset of **UniRef50** (Suzek et al., 2007), containing 72.1M representative sequences clustered at 50% identity. For evaluation, we select 6 tasks from the **PEER** benchmark (Xu et al., 2022), covering protein function, structure, and interactions. Detailed descriptions are provided in Appendix A.1.2.

### 3.2 ARCHITECTURE SPACE DESIGN AND SUPERNET CONSTRUCTION

In this section, we introduce the design to build a vast and diverse search space and the construction of supernet for efficient training process.

**Basic Blocks** We begin by introducing the basic modules defined as $\mathcal{M}$, which are derived from a variety of architectures widely used for biological data such as **CNN** (Krizhevsky et al., 2012), **LSTM** (Hochreiter & Schmidhuber, 1997), **Transformer** (Vaswani et al., 2017), **Mamba** (Gu & Dao, 2024) and **Hyena** (Poli et al., 2023). The diversity in block types ensures our search can cover different inductive biases. Details of these modules could be found in Appendix A.2.

**Paths formed by blocks** Each path $a \in \mathcal{A}$ is a sequence of basic modules $(l_1, l_2, \ldots, l_d)$, where its structure is determined by three variations:

1. *Network Depth* (**d**): The number of blocks in a path, chosen from a predefined set of possible depths, $\mathcal{D} = \{D_{\min}, \ldots, D_{\max}\}$.

2. *Module Type* (**m**): A tuple $\mathbf{m} = (m_1, m_2, \ldots, m_d)$ of length $d$, where each element $m_i \in \mathcal{M}$ specifies the module for the $i$-th layer.

3. *Hidden Dimension* (**h**): A tuple $\mathbf{h} = (h_1, h_2, \ldots, h_d)$ of length $d$, where each $h_i$ is a hidden dimension selected from a set of possible widths $\mathcal{H}$.

A complete path $a$ is constructed from a tuple $(\mathbf{h}, \mathbf{m})$ of length $d$. For $i \in \{1, \ldots, d\}$, the $i$-th layer $l_i$ is of type $m_i$ and maps an input of dimension $h_{i-1}$ to an output of dimension $h_i$. The initial dimension $h_0$ is the embedding dimension which is a fixed hyperparameter.

**Search Space Design** Let $\mathcal{C}_{\text{dim}}^{(d)}$ and $\mathcal{C}_{\text{type}}^{(d)}$ be the sets of all permissible dimension and module-type configurations for a given depth $d$, respectively. The total search space $\mathcal{A}$ is then the union of the Cartesian products of these configuration sets across all possible depths:

$$\mathcal{A} = \bigcup_{d \in \mathcal{D}} \left( \mathcal{C}_{\text{dim}}^{(d)} \times \mathcal{C}_{\text{type}}^{(d)} \right) \tag{1}$$

This formulation programmatically generates a vast yet structured set of candidate architectures, balancing model depth, width, and module choice. However, as architectural depth increases, the number of combinations grows rapidly. To control the search space size, we used three pruning methods, including rule-based pruning, distance-based pruning, and clustering, to select representative paths. By pruning, we reduce the number of combinations from 67 million to 360. Further details are provided in Appendix A.3.

**Supernet Construction** Training each path $a \in \mathcal{A}$ from scratch is computationally expensive. To overcome this, we adopt a supernet (Bender et al., 2018) approach using weight sharing. The key insight is that many architectures within $\mathcal{A}$ reuse identical building blocks. For example, a specific Transformer block with a 256 input dimension and a 512 output dimension is a component in different paths. Instead of creating and training a new instance of this block for each path, we initialize it only once and share its weights across all paths that use it. Therefore, the supernet is a single and large network that effectively contains all candidate architectures as paths. First, we identify the set of all unique blocks used in paths across the entire search space:

$$L = \{l \mid \exists a \in \mathcal{A}, l \in a\} \tag{2}$$

Each unique module is defined by its type (e.g., CNN, Mamba) and its dimensions. Other hyperparameters (e.g., CNN kernel sizes, Transformer attention heads) are fixed. The supernet's single global weight set $W$ is the union of all unique block weights:

$$W = \bigcup_{l \in L} W_l \tag{3}$$

where $W_l$ are the trainable parameters for a unique layer $l$. The weights for any specific path denoted as $w(a)$ are a subset of these shared weights:

$$w(a) = \{W_{l_i} \mid l_i \in a\} \subseteq W \tag{4}$$

### 3.3 SUPERNET PRETRAINING

**One-Shot Supernet Training** We adopt a random path sampling mechanism inspired by the Single Path One-Shot approach (Guo et al., 2020). In each forward pass, we activate a single path

through the supernet. We sample from a uniform distribution to select which path to activate. This paradigm decouples the training of different paths and ensures every operation is trained equally, enabling a more accurate and stable performance evaluation.

**Self-Supervised Pretraining Objectives** The supernet denoted $\mathcal{A}$ is trained by stochastically sampling architectures from the search space. In each training step, a path $a$ is sampled, and the supernet's shared weights $W$ are updated by minimizing self-supervised loss $\mathcal{L}$. The overall training objective is to optimize the expectation of this loss over the distribution of all possible architectures:

$$\min_{W} \mathbb{E}_{a \sim \mathcal{A}}[\mathcal{L}(\mathcal{A}(X; w(a)))] \tag{5}$$

where $X$ is the input data, $a \sim \mathcal{A}$ denotes a path sampled (typically uniformly) from the search space $\mathcal{A}$, and $w(a) \subset W$ are the weights corresponding to the block in path $a$.

To ensure the supernet learns meaningful and generalizable features, we pretrain it with three commonly used self-supervised learning (SSL) objectives: Masked Modeling (MM), Contrastive Learning (CL), and Next Token Prediction (NTP). This pretraining step is performed on a large corpus of unlabeled biological data before the supernet is used for architecture evaluation on specific downstream tasks. Details of these training objectives could be found in Appendix A.4.

**Tokenization method** The tokenizer is a critical component in converting sequence data into embeddings for the model. To thoroughly investigate the impact of tokenization on performance, we conducted experiments using different tokenizers, including those from the k-mer and Byte Pair Encoding (BPE) (Sennrich et al., 2016). *K-mer tokenizers* operate by dividing a sequence into overlapping or non-overlapping substrings of a fixed length, denoted as $k$. **BPE** is a subword tokenization strategy that begins with a vocabulary of individual characters and iteratively merges the most frequently occurring adjacent pairs of tokens into a new, single token.

### 3.4 ARCHITECTURE EVALUATION AND RANKING

**Evaluation Protocol** To fairly assess the performance of sampled architectures, **we optimize each path independently, rather than fine-tuning the entire supernet as a whole.** This individual optimization is crucial because the supernet training strategy (via stochastic sampling) is specifically designed to prevent **co-adaptation** between modules Bender et al. (2018); Guo et al. (2020). Consequently, the resulting shared parameters are optimized for the **expected performance** over the entire search space, representing a **compromise** across varying topologies rather than being optimal for any single path. Therefore, evaluating a specific architecture requires adapting these weights to its unique structure (or training from scratch) to reveal its true potential. We conduct this independent assessment using two paradigms distinct by their weight initialization:

- **Pretrain and finetune:** The parameters of the sampled architecture $a$, denoted as $w(a)$, are initialized by inheriting the corresponding weights from the pretrained supernet $W$. The architecture is then fine-tuned to minimize the task-specific loss:

$$\min_{w(a)} \mathcal{L}_{\text{task}}(\mathcal{A}(X_{\text{task}}; w(a))), \quad \text{s.t.} \quad w(a)_{\text{init}} \subset W. \tag{6}$$

- **Training from Scratch:** The parameters $w'(a)$ are re-initialized randomly (independent of the supernet $W$) and trained from scratch on the downstream task. The optimization objective remains the same, but the optimization trajectory starts from a random state:

$$\min_{w'(a)} \mathcal{L}_{\text{task}}(\mathcal{A}(X_{\text{task}}; w'(a))), \quad \text{s.t.} \quad w'(a)_{\text{init}} \sim \mathcal{D}_{\text{random}} \tag{7}$$

**Architecture Ranking Strategy** We rank the sampled architectures $\mathbb{S}$ based on their aggregated performance across the set of downstream tasks $\mathcal{T}$. Since tasks utilize different metrics with varying scales and statistical variances (e.g., Accuracy vs. RMSE), a direct summation would disproportionately favor tasks with wider numerical distributions. To address this heterogeneity and ensure that each task contributes equally to the final ranking, we employ Z-score normalization to standardize

the performance metrics before aggregation. We define the unified score for an architecture $a$ as the mean of these standardized metrics:

$$\text{Score}(a) = \frac{1}{|\mathcal{T}|} \sum_{t \in \mathcal{T}} s_t \cdot \frac{P_t(a) - \mu_t}{\sigma_t}, \tag{8}$$

where $P_t(a)$ denotes the raw performance of architecture $a$ on task $t$. The terms $\mu_t$ and $\sigma_t$ represent the mean and standard deviation of the performance scores for task $t$ across all sampled candidates. The coefficient $s_t \in \{1, -1\}$ aligns the optimization direction, taking $1$ for metrics where higher is better (e.g., Accuracy) and $-1$ for error-based metrics (e.g., RMSE). Finally, the top candidates $\mathbb{A}_{\text{top}}$ are identified by selecting the $k$ architectures with the highest $\text{Score}(a)$.

**Optimal Architecture as Foundation Model** Once the top one ranking architecture $a^* \in \mathcal{A}$ is identified, we randomly initialize its parameters, denoted as $\theta$, and pretrain it:

$$\min_\theta \mathcal{L}(a^*(X; \theta)). \tag{9}$$

To evaluate the foundation model's transferability, we subsequently fine-tune the pretrained parameters $\theta$ on each specific downstream task $t$:

$$\min_w \mathcal{L}_{\text{task}}(\mathcal{A}(X_{\text{task}}; w)), \quad \text{s.t.} \quad w_{\text{init}} \leftarrow \theta \tag{10}$$

### 3.5 ARCHITECTURE PREDICTION METHOD

Training every candidate in a vast BIOARC search space is computationally expensive. Leveraging our observation that **optimal architectures for functionally similar tasks exhibit significant topological similarity**, we propose a hierarchy of three approaches, ranging from numerical regression to high-level semantic reasoning agents, to efficiently exploit BioArc explored (architecture, task, performance) tuples for identifying optimal architectures for unseen biological tasks. We provide empirical verification of the observation in Section 4.2 and Appendix A.6.5.

**Neural Network Prediction** We design a neural predictor denoted as $\mathcal{P}$ that maps an architecture-task pair $(a, t)$ to a predicted performance score. The model takes an architecture embedding $\mathbf{h}_a$ and a task embedding $\mathbf{h}_t$ as input and is trained by minimizing the Mean Squared Error against the ground-truth performance $y_{a,t}$. This objective is formally expressed through the loss function $\mathcal{L}(\mathcal{P})$:

$$\mathcal{L}(\mathcal{P}) = \mathbb{E}_{(a,t) \sim \mathcal{A}} \left[ (\mathcal{P}(\mathbf{h}_a, \mathbf{h}_t) - y_{a,t})^2 \right] \tag{11}$$

Following previous work (Ma et al., 2019; White et al., 2021), we represent each architecture $a$ as a graph with its node features $\mathbf{h}_a$ formed by concatenating of its one-hot encoded module type $\text{O}(m_i)$ and the normalized dimensions of its input and output, $Z(h_{i-1})$ and $Z(h_i)$. For task embeddings, a pretrained language model (PLM) is used to encode the task's textual description, $d_{\text{task}}$, into a vector $\mathbf{h}_t$. These encoding processes are defined as:

$$\mathbf{X}_a = (\text{O}(m_i) \| Z(h_{i-1}) \| Z(h_i))_{i=1}^d, \qquad \mathbf{h}_a = \mathbf{GNN}(\mathbf{I}, \mathbf{X}_a), \qquad \mathbf{h}_t = \text{PLM}(d_{\text{task}}) \tag{12}$$

where $d$ is the depth of the architecture. An example of architecture embedding is in Appendix A.5.

**LLM + RAG** Moving beyond numerical regression, we leverage LLMs augmented with retrieval capabilities. To provide relevant context, we first encode the input task description into **an embedding vector and retrieve the top-$n$ most similar historical tasks** based on vector similarity. These retrieved tasks, along with their corresponding top-$k$ architectures, are formatted as a **knowledge base** and **performance records** within the prompt. The LLM is then instructed to act as an analyst: it first evaluates the new task's characteristics, identifies the most relevant matches from the retrieved knowledge base, and predict the top-$m$ architecture. This approach enables the model to explicitly reason about task relationships and empirical performance. $n, k, m$ are hyperparameters.

**BIOARC Agent** To address the limitations of simple embedding-based retrieval in capturing biological nuances, we introduce BIOARC Agent. As detailed in Table 1, this system transforms unstructured queries into empirically grounded predictions. Specifically, the system identifies the top-$k$ most semantically similar historical tasks and retrieves their corresponding top-$n$ high-performing architectures. Then Predictor conduct reasoning and output top-$m$ architectures. $n, k, m$ are hyperparameters. Full prompts and operational details are provided in Appendix A.9.

Table 1: Functional breakdown of the BIOARC Agent pipeline. The system progresses from parsing raw text to synthesizing an optimal design.

| Role | Input | Output |
|------|-------|--------|
| **Analyst** | Raw user task description | Structured metadata (e.g., modality, objective) |
| **Task Retriever** | Structured metadata | Semantically aligned tasks in Knowledge Base |
| **Arch. Retriever** | Aligned tasks | Proven architectures & empirical performance |
| **Predictor** | Retrieved architectures & metrics | Predict optimal architecture design |

**Architecture Prediction Evaluation** To strictly validate our prediction methods, we utilize the comprehensive performance data from Section 3.3 as ground truth. We establish two distinct settings to test the predictor's cognitive capabilities: the first is **Supervised Setting**, where datasets within the same task are split into known and unseen sets (e.g., TFP-0 and TFP-1 are known, while TFP-2 is unseen); and the second is **Transfer Setting**, where the model is evaluated on novel tasks disjoint from the known set (e.g., PD is known and SSP is unseen) to verify its capacity for first-principles reasoning beyond simple memorization. We use each method to predict top-$K$ architectures and quantify the alignment between predictions and this ground truth using Precision@$k$, Recall@$k$, and Hit Rate@$k$. Detailed mathematical definitions are provided in Appendix A.6.9

## 4 EXPERIMENT

Having detailed our construction of BIOARC framework, we aim to answer the following research questions with our experiments. **RQ1:** How competitive are the BIOARC architectures against large-scale foundation models when initialized from the pretrained supernet and finetuned? Moreover, how do they perform when trained from scratch? **RQ2:** What are the common architectural properties that characterize high-performing models? **RQ3:** Does large-scale pretraining on our optimal architecture yield a foundation model that outperforms existing biological foundation models? **RQ4:** What is the effect of different training strategies? **RQ5:** What is the effect of different tokenizers? **RQ6:** How effective are different methods for optimal architecture prediction?

### 4.1 SETTINGS AND BASELINES

**Baselines** For DNA, we use Nucleotide Transformer (Dalla-Torre et al., 2023), DNABERT-2 (Zhou et al., 2024), and VQDNA (Li et al., 2024) as baselines. These pretrained large foundation models all share human DNA GRCh38 in pretraining with additional multi-species genome datasets. For protein, we use ESM-1 and ESM-2 (Rives et al., 2021; Lin et al., 2023) and ProtBert (Elnaggar et al., 2021) as baselines. ProtBert is pretrained on BFD (Steinegger et al., 2019) with 393B amino acids and ESM-1b is pretrained on UniParc (Consortium, 2007) with 86B amino acids.

**Our settings** We use human DNA GRCh38 for DNA pretraining and 10% of randomly sampled UniRef50 (Suzek et al., 2007) for protein pretraining. Details of pretraining, finetune, and architecture prediction settings could be found in Appendix A.6.1.

### 4.2 MAIN RESULTS

In this section, we wish to answer RQ1, RQ2 and RQ3. **For RQ1**, we observe the following. **First, architectures discovered by BIOARC are highly competitive, often outperforming larger, pretrained models with much smaller model size.** As detailed in Table 2, our models achieve superior performance across all DNA-related tasks, in some cases by a substantial margin. For the protein-centric tasks in Table 3, our results reveal a clear divide. BIOARC-discovered models excel on sequence-level tasks like Solubility and HumanPPI, outperforming larger counterparts. We attribute our success on sequence-function tasks to task-specific inductive bias: our hybrid topologies adapt to the intrinsic signal density, maximizing data efficiency. Conversely, the gap on structural tasks stems from Pretraining Scale and Spatial Inductive Bias (further discussed in RQ2). Structural inference requires vast evolutionary memorization, which is limited by our subset training. Furthermore,

Table 2: Performance on different DNA tasks. **Bold** and underline indicate the best and second-best result respectively. We report the average size of top-performing architectures across tasks. Definition of variants could be found in Section 3.3 and Appendix A.6.1.

| Method | #Param | #Data | TFP | | | | | PD | | | CPD | | | SSP |
|---|---|---|---|---|---|---|---|---|---|---|---|---|---|---|
| | | | 0 | 1 | 2 | 3 | 4 | all | notata | tata | all | notata | tata | Reconstruction |
| HyenaDNA (one-hot) | 6.6M | 3.1B | 62.30 | 67.86 | 46.85 | 41.78 | 61.23 | 47.38 | 52.24 | 5.34 | 36.95 | 35.38 | 72.87 | 72.67 |
| NT-2500M-1000g (6-mer) | 2500M | 20.5T | 66.31 | 68.30 | 58.70 | 49.08 | 67.59 | 90.95 | 93.07 | 75.80 | 67.39 | 67.46 | 69.66 | 85.78 |
| DNABERT-2 (BPE) | 117M | 262B | 71.99 | 76.06 | 66.52 | 58.54 | 77.43 | 86.77 | 94.27 | 71.59 | 69.37 | 68.04 | 74.17 | 84.99 |
| VQDNA (HRQ) | 103M | 262B | 72.48 | 76.43 | 66.85 | 58.92 | 78.10 | 90.75 | 94.48 | 74.52 | 71.02 | 70.58 | 78.50 | 89.53 |
| BIOARC (only-ft) | 4.55M | 0 | 84.70 | 85.90 | 86.20 | **77.70** | **90.10** | **94.38** | 96.46 | **84.34** | **83.87** | 84.51 | **90.05** | **91.17** |
| BIOARC (mask-ft) | 4.89M | 3.1B | **84.80** | 86.00 | 85.80 | 77.10 | 89.20 | 92.62 | **96.55** | 83.20 | 83.60 | **85.43** | 89.40 | 90.84 |
| BIOARC (con-ft) | 3.28M | 3.1B | **84.80** | **86.10** | **86.50** | 77.50 | 89.30 | 93.12 | 96.25 | 83.69 | 83.53 | 84.66 | **90.05** | 91.06 |
| BIOARC (ntp-ft) | 6.58M | 3.1B | 84.20 | 85.90 | 82.80 | 75.50 | 89.10 | 91.82 | 96.25 | 76.02 | 82.26 | 84.38 | 74.55 | 83.76 |

Table 3: Performance on different protein tasks. **Bold** and underline indicate the best and second-best result respectively. We report the average size of top-performing architectures across tasks. Definition of variants could be found in Section 3.3 and Appendix A.6.1.

| Method | #Param | #Data | Solubility | HumanPPI | PPIAffinity | Fold | Subcellular | Binary |
|---|---|---|---|---|---|---|---|---|
| ProtBert | 419.9M | 393B | 68.15 | 77.32 | 2.195 | 16.94 | 76.53 | 91.32 |
| ESM-1b | 652.4M | 86B | 70.23 | 78.17 | 2.281 | 28.17 | **78.13** | 92.40 |
| ESM2-1b | 652.4M | 86B | 74.13 | 80.49 | **2.134** | **32.31** | 80.45 | **92.79** |
| ESM2-8m | 8M | 86B | 73.48 | 80.16 | 3.098 | 22.14 | 71.47 | 91.25 |
| BIOARC (only-ft) | 2.88M | 0B | 73.34 | **80.59** | 2.330 | 15.88 | 68.48 | 89.25 |
| BIOARC (mask-ft) | 3.14M | 2.1B | 74.14 | 74.26 | 2.210 | 15.88 | 68.52 | 88.45 |
| BIOARC (con-ft) | 2.49M | 2.1B | **74.24** | 77.22 | 2.760 | 15.74 | 68.34 | 88.05 |

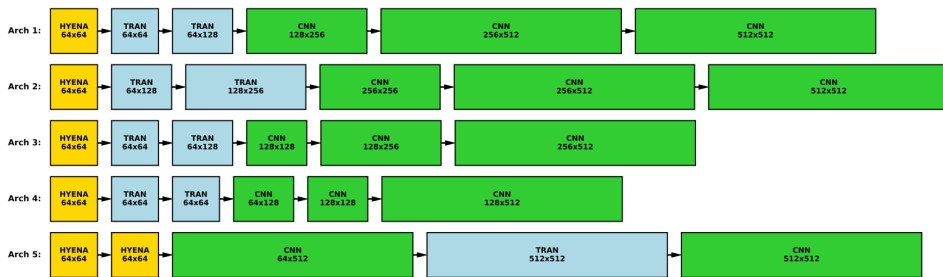

Figure 2: This figure illustrates the top five performing DNA model architectures (Arch 1-5), identified by averaging their performance across the different tasks on which they were directly trained from scratch. These architectures are composed of combinations of HYENA, Transformer and CNN modules. It is observable that these high-performing architectures shares a common pattern. Results on protein is shown in Appendix A.6.4.

structure prediction relies on relative distances. The superiority of the small ESM-2 over the larger ProtBERT highlights the necessity of Rotary Positional Embeddings (RoPE) (Su et al., 2023). Our reliance on absolute positioning lacks this spatial prior, creating a performance ceiling. **This highlights a key principle for biology model design that optimal architecture depends on the nature of the task.** For those demanding intricate feature extraction from the sequence itself, a specialized architecture is key. For others that rely on vast, implicit priors learned from data at scale, massive pretraining is essential.

**For RQ2**, we observe three key findings. **First, top-performing architectures converge on specific structural patterns.** As shown in Figure 2, optimal models typically initiate with Hyena block to first captures long-range dependencies, with Transformer blocks in the middle to model complex contextual relationships and CNN blocks at the end to extract critical features. **Second, while optimal architectures are task-specific, they exhibit significant commonalities within task families.** For instance, the top 10% architectures for Transcription Factor Prediction-3 and 4 share 96.0% similarity (Figure 7, Appendix A.6.5). **Third, hybrid architectures consistently outperform single-module baselines,** validating the effectiveness of our search space design (see Appendix A.6.6).

**To answer RQ3**, we selected a overall top-performing BIOARC architecture and applied the same large-scale pretraining regimen used for established DNABERT2 model (Zhou et al., 2024). As shown in Figure 3, **with only 1/20 model size and 1/10 training steps, BIOARC-based foundation model outperforms the human-designed architectures on downstream DNA tasks.** This result,

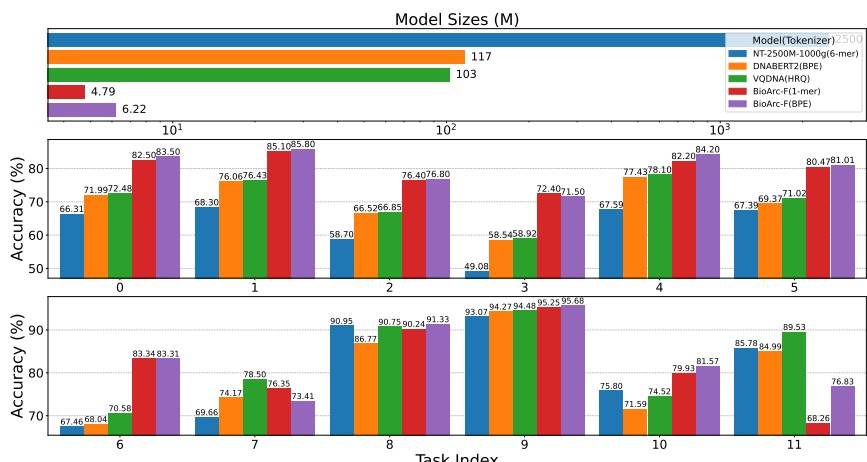

Figure 3: Performance of BIOARC-Discovered Architecture as a Foundation Model Backbone, noted as **BIOARC-F**. The architecture is selected by the top average performance across all tasks and pretrained for 1/10 training steps of baselines.

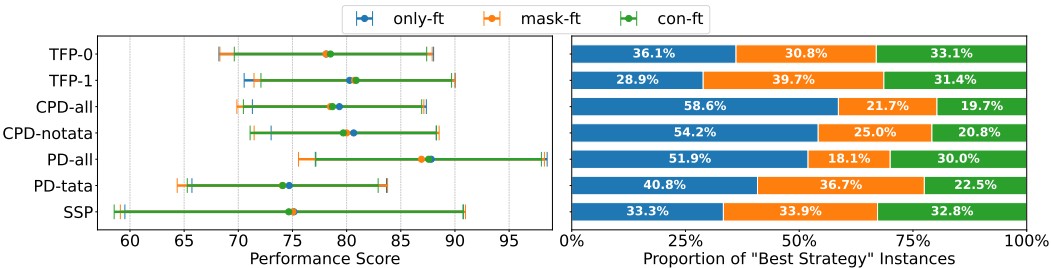

Figure 4: Performance of different training strategies on DNA. In the left panel, each cell shows the mean performance ± standard deviation across all architectures. In the right panel, each bar shows the percentage of the total 360 architectures that chose that training strategy to yield the best performance. More results on Protein could be found in Appendix A.6.8.

combined with our findings for RQ1, indicates that BIOARC is highly effective for identifying both high-performing specialized architectures and robust backbones for foundational models in biology.

### 4.3 STUDY ON TRAINING STRATEGY

**To answer RQ4**, we summarize the training strategy results in Figure 4. **First, pretraining does not guarantee gains**. Training from scratch (only-ft) achieves the highest win rates on CPD and PD tasks, with only marginal differences elsewhere. **Secondly, no pretraining strategy dominates**. Masked modeling and contrastive learning show no consistent performance gap, yielding comparable average scores across tasks. **Thirdly, pretraining a universal architecture achieves performance comparable to task-specific optimization.** BIOARC-based foundation model shown in Figure 3 achieves similar performance as the best individual architectures found for downstream task. This indicates that our architecture search can find some universal best architecture for foundation model building for each data modality that competes with the best task-specific architectures selected individually for each specific task.

### 4.4 STUDY ON TOKENIZATION

In this section, **we address RQ5** by analyzing the impact of different tokenization methods. **First, we find that the optimal tokenizer choice is highly architecture-dependent.** As shown in Fig-

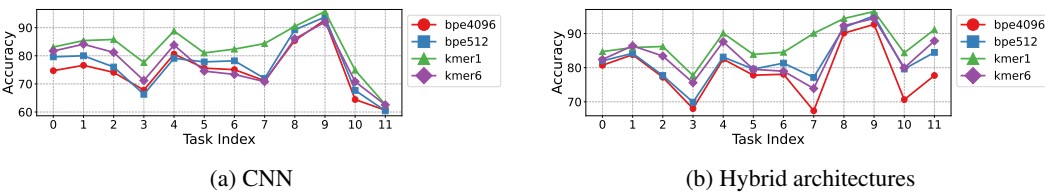

(a) CNN                                    (b) Hybrid architectures

Figure 5: Effect of different tokenization on various architectures training from scratch on the DNA tasks. More results could be found in Appendix A.6.9.

ure 5, the Transformer-based architecture performs best with a 6-mer tokenizer, while the CNN-based model achieves its highest scores with a 1-mer tokenizer. This reveals a deep interplay between architecture and tokenization, demanding their co-optimization. **Second, the tokenizer's effectiveness interacts with the training strategy.** As shown in Figure 3, BIOARC-F with BPE tokenizer outperforms 1-mer tokenizer, which contradicts the situation in task-specific models. This suggests that pretraining propels complex tokenization, while simple tokenization methods are less reliant on pretraining. More comprehensive results are available in the Appendix A.6.9.

### 4.5 ARCHITECTURE PREDICTION

In this section, **we address RQ6** comparing the prediction effectiveness of different methods. Based on our observations from RQ2, we conducted experiments in both supervised and transfer settings with detailed data split available in Appendix A.6.1. We use GPT-4o as default backbone and compared the performance of multiple LLM models. From Table 4, **we find that that BIOARC Agent consistently outperforms other methods in both settings.** We attribute this improvement to the modular multi-agent design, which decouples the decision-making process to minimize hallucination and error propagation. Furthermore, unlike static embeddings, our natural language-driven retrieval ensures a semantic matching of tasks, allowing the system to identify historical precedents that share genuine structural and functional requirements. We evaluate different LLM models, results could be found in Appendix A.6.9.

Table 4: Architecture prediction methods comparison under supervised and transfer setting.

| Setting | Metric | NN Predictor | | | | | LLM+RAG | | | BIOARC Agent | | |
|---------|--------|------|------|------|------|------|------|------|------|-------|-------|-------|
| | | @5 | @10 | @15 | @20 | @30 | @1 | @3 | @5 | @1 | @3 | @5 |
| | Hit Rate | 0.167 | 0.333 | 0.333 | 0.333 | 0.667 | 0.000 | 0.167 | 0.167 | **0.500** | **0.500** | **0.500** |
| **Supervised** | Precision | 0.033 | 0.033 | 0.022 | 0.017 | 0.028 | 0.000 | 0.056 | 0.111 | **0.500** | **0.444** | **0.300** |
| | Recall | 0.033 | 0.067 | 0.067 | 0.100 | 0.167 | 0.000 | 0.056 | 0.067 | **0.100** | **0.267** | **0.300** |
| | Hit Rate | 0.000 | 0.250 | 0.250 | 0.250 | 0.500 | 0.000 | 0.000 | 0.000 | **0.250** | **0.250** | **0.500** |
| **Transfer** | Precision | 0.000 | 0.025 | 0.017 | 0.013 | 0.017 | 0.000 | 0.000 | 0.000 | **0.250** | **0.250** | **0.300** |
| | Recall | 0.000 | 0.050 | 0.050 | 0.050 | 0.100 | 0.000 | 0.000 | 0.000 | **0.050** | **0.150** | **0.300** |

## 5 CONCLUSION

In this work, we introduce BIOARC, a novel framework that is based on neural architecture search. We systematically explored a vast architecture space and identified new, high-performing foundational and task-specific architectures, often outperforming much larger established pretrained models on various tasks. We also conducted a systematic evaluation of how tokenization and training strategies interact with model architecture, clarifying the critical impact of these choices. We developed a BIOARC Agent system that can efficiently predict optimal architectures for biological tasks. Ultimately, BIOARC provides a foundational resource and a principled methodology to aid in creating the next generation of models tailored for biology.

## 6 REPRODUCIBILITY STATEMENT

We provide a partial, anonymized implementation of our method at at ***BioArc-9794*** for review purposes. To ensure full reproducibility, we have detailed all experimental settings in the Appendix. This covers the model architectures, hyperparameter choices, training processes used in our work. All the dataset used in our work in public.

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

# A APPENDIX

## A.1 BENCHMARK DETAILS

### A.1.1 DNA

**Transcription Factor Prediction (TFP, Task 0-4):** This task involves predicting transcription factor binding sites (TFBS) in the human genome using data derived from 690 ENCODE ChIP-seq experiments.

**Core Promoter Detection (CPD, Task 5-7):** This task involves identifying the precise location of core promoter regions, which are essential for initiating gene transcription, focusing on predicting the core promoter region only, the central region closest to the TSS and start codon. A much shorter context window (center -34 +35 bp around TSS) is provided, making this a more challenging task than proximal promoter prediction.

**Promoter Detection (PD, Task 8-10):** This task requires the model to identify human proximal promoter regions, which contain primary regulatory elements crucial for transcription initiation. We construct three dataset variations: TATA, non-TATA, and a combined All set. Positive samples are extracted from the Eukaryotic Promoter Database (EPDnew) (Dreos et al., 2013), covering the window from -249 to +50 bp relative to the Transcription Start Site (TSS). Negative control strategies differ by subset: the TATA dataset uses random non-promoter genomic sequences containing TATA motifs, while the non-TATA dataset utilizes randomly substituted sequences to represent the null class.

**Splice Site Prediction (SSP, Task 11):** This task focuses on splice site prediction in the human genome, a process vital for understanding protein diversity and genetic diseases.

### A.1.2 PROTEIN

**Solubility Prediction (Task 12):** a binary classification task that determines if a protein is soluble or insoluble. To ensure models can generalize to new and dissimilar proteins, the dataset is split so that the training proteins have less than 30% sequence identity with the test proteins. This is a critical task because good solubility is an essential property for functional proteins, especially in pharmaceutical research and industry. The ultimate goal is to drive the development of more effective computational tools that can predict a protein's solubility based solely on its amino acid sequence.

**Human PPI Prediction (Task 13):** A binary classification task to determine whether a pair of human proteins will physically interact. The data is split into an 8:1:1 ratio for training, validation, and testing, and the task is designed to test how well models can generalize to dissimilar protein sequences. The work is significant because understanding the human protein interactome is vital for deciphering disease mechanisms. This task serves as a benchmark to drive the development of more effective machine learning models for PPI prediction.

**Fold Classification (Task 14):** A classification task that predicts the overall 3D structural shape (fold) of a protein from 1,195 possible categories. The model is specifically tested on its ability to perform remote homology detection, meaning it must recognize proteins with similar structures even if their sequences are very different. To achieve this, entire superfamilies of proteins are withheld from the training data and used only for testing. The goal is to automate fold classification using protein sequences, which is important for drug design and functional analysis because most known protein structures have not yet been manually classified.

**PPI Affinity Prediction (Task 15):** A regression task that estimates the binding strength between two proteins. Using the SKEMPI dataset, the data is split based on the number of mutations to test the model's generalization capabilities in a scenario mimicking multi-round protein engineering: Training set is wild-type proteins and mutants with less then 2 mutations. Validation set is mutants with 3 or 4 mutations. Test set is mutants with more then 4 mutations. The primary impact of this task is its direct application to protein binder design, where accurately predicting the binding strength of candidate molecules is crucial for developing new therapeutics and biotechnologies.

**Subcellular Location Prediction (Task 16):** A multiclass classification task involves predicting which of 10 possible locations a protein resides in within a cell. It is a multi-class classification problem where models are trained and tested using the DeepLoc dataset, which is specifically partitioned to evaluate performance on homologous proteins. The primary impact of this research is in drug discovery. Knowing a protein's location helps identify it as a potential drug target, and an accurate, high-throughput prediction tool can significantly accelerate this process.

**Binary Location Prediction (Task 17):** A simplified version of subcellular localization, framed as a binary classification problem designed to classify proteins as either membrane-bound or soluble, using a binary label like 0 or 1. The model is trained and tested on data from the DeepLoc dataset, with a key evaluation being its ability to generalize and make accurate predictions for homologous (structurally similar) proteins. The task is significant because it helps efficiently distinguish between soluble proteins, which are free-floating, and membrane-bound proteins, which are attached to cell membranes and can have important catalytic functions.

## A.2 MODULES DESIGNS

In this section, we provide the detailed architectural specifications for the various building blocks employed in our framework. Table 5 presents a summary of these designs, outlining the core layers, normalization and activation mechanisms, as well as the specific hyperparameter settings (e.g., kernel sizes, initialization schemes) used in our experiments.

Table 5: Detailed Specifications of Architectural Modules

| Module | Core Layer | Norm / Activation | Key Hyperparameters |
|---|---|---|---|
| CNN | Conv1d ($D_{in} \rightarrow D_{out}$) | ReLU $\rightarrow$ BatchNorm1d | Kernel: $\{5, 9\}$, Padding: $\{2, 4\}$ |
| Hyena | HyenaEncoderLayer | *None* (Identity) | Proj: Linear ($D_{in} \rightarrow D_{out}$) optional; Model Dim: $D_{out}$ |
| Transformer | TransformerEncoderLayer | LayerNorm (internal) | Heads: $D_{out}/64$; Input matched to $D_{out}$ |
| Mamba | Mamba Layer | LayerNorm (pre-block) | Proj: Linear ($D_{in} \rightarrow D_{out}$) optional |
| LSTM | LSTM (1 layer) | Sigmoid/Tanh (internal gates) | Hidden: $D_{out}$, Dropout: 0.4, Init: Orthogonal (rec), Xavier (in), Bias 0 |

## A.3 SEARCH SPACE PRUNING

The combinatorial nature of search space design, while comprehensive, leads to an exponential growth in the number of candidate architectures as the number of choices increases. To maintain a computationally tractable search space without sacrificing diversity, we employ three strategies to select a representative subset of configurations for both module types ($\mathcal{C}_{\text{type}}^{(d)}$) and hidden dimensions ($\mathcal{C}_{\text{dim}}^{(d)}$) at each depth $d \in \mathcal{D}$ (in Equation 1).

**Rule-based pruning.** We enforces a monotonic non-decreasing width constraint (i.e., $h_i \geq h_{i-1}$ for $i \geq 1$) to encourage progressive feature extraction.

**Distance-based pruning.** We first generate all possible valid dimension paths that satisfy the non-decreasing constraint. To ensure diversity from the outset, we employ a greedy selection algorithm. A candidate path is only added to our initial representative set if it is sufficiently distant from all previously selected paths. The distance between two dimension paths, $\mathbf{h}_a$ and $\mathbf{h}_b$, is measured as the Euclidean distance in a **log-transformed space**. Specifically, given full paths $\mathbf{h}'_a = (h_0, h_{1,a}, \ldots, h_{d,a})$ and $\mathbf{h}'_b = (h_0, h_{1,b}, \ldots, h_{d,b})$, where $h_0$ is the fixed embedding dimension, their distance is:

$$D(\mathbf{h}'_a, \mathbf{h}'_b) = \sqrt{\sum_{i=0}^{d} \left( \log_2(h_{i,a}) - \log_2(h_{i,b}) \right)^2}$$

A candidate path $\mathbf{h}_{\text{cand}}$ is kept only if $D(\mathbf{h}'_{\text{cand}}, \mathbf{h}'_{\text{rep}}) \geq \tau$ for all paths $\mathbf{h}_{\text{rep}}$ already in the representative set, where $\tau$ is a predefined distance threshold. Using a log scale ensures that the selection is sensitive to relative changes in dimension (e.g., distinguishing between 64 and 128) rather than absolute differences.

**Cluster-based pruning.** We deploy clustering method in both module types and hidden dimension orders. For a given depth $d$, the total number of possible module type configurations is $|\mathcal{M}|^d$. To reduce this number to a manageable target $k_d$, we perform clustering to identify a set of diverse and representative paths. The procedure is as follows:

1. **Vector Representation**: Each module type path $\mathbf{m} = (m_1, m_2, \ldots, m_d)$ is first transformed into a numerical vector. We apply **one-hot encoding** to each module $m_i \in \mathcal{M}$, converting the categorical sequence into a high-dimensional numerical representation. This results in a flattened vector for each path.

2. **K-Means Clustering**: We then apply the K-Means clustering algorithm to the set of all one-hot encoded vectors. The number of clusters is set to the desired number of representative configurations $k_d$. K-Means partitions the paths into $k_d$ clusters by minimizing the within-cluster sum of squares.

3. **Centroid-based Selection**: The centroid of each cluster represents the mean of the paths within it. Since a centroid may not correspond to a valid, discrete module type path, we select the actual path from the dataset that is closest to each cluster centroid in terms of Euclidean distance. This ensures that our final set $\mathcal{C}_{\text{type}}^{(d)}$ consists of $k_d$ valid and diverse module type configurations.

The search space for hidden dimension paths $\mathbf{h} = (h_1, h_2, \ldots, h_d)$ is also vast. We apply the same pipeline on it, except changing the first encoding step.

By applying these structured reduction techniques to both the module type and hidden dimension configurations, we construct a final search space $\mathcal{A}$ that is both diverse and computationally feasible, allowing for an efficient yet comprehensive exploration of the architectural landscape.

### A.4 TRAINING STRATEGIES

**Masked Modeling (MM)** For a given input sequence $S = (x_1, x_2, \ldots, x_L)$, we randomly mask a fraction of its tokens to create a corrupted sequence $\tilde{S}$. The supernet is then tasked with predicting the original tokens for the masked positions based on the contextual information provided by the unmasked tokens. This process compels the model to learn intricate local dependencies and contextual relationships within biological sequences.

$$\mathcal{L}_{\text{MM}} = - \sum_{t \in \mathcal{M}} \log p(x_t \mid \tilde{S})$$

where $\mathcal{M}$ is the set of indices of the masked tokens.

**Contrastive Learning (CL)** This approach aims to learn an embedding space where representations of similar sequences are pulled closer together, while those of dissimilar sequences are pushed apart. For a batch of sequences $\{S_i\}_{i=1}^B$, we generate a "positive" pair for each sequence $S_i$ by applying data augmentations, resulting in encoded representations $z_i$ and $z_j$. All other sequences in the batch are treated as "negative" examples. The model is trained to maximize the cosine similarity of positive pairs while minimizing it for negative pairs:

$$\mathcal{L}_{\text{CL}} = - \log \frac{\exp(\text{sim}(z_i, z_j)/\tau)}{\sum_{k \neq i} \exp(\text{sim}(z_i, z_k)/\tau)}$$

where $\text{sim}(\cdot)$ denotes cosine similarity, and $\tau$ is a temperature hyperparameter.

**Next Token Prediction (NTP)** For a given input sequence $S = (x_1, x_2, \ldots, x_L)$, this objective leverages the autoregressive nature of biological sequences. The supernet is tasked with predicting

the next token $x_{t+1}$ based on the preceding context in $S$. This process compels the model to learn the sequential grammar and causal dependencies:

$$\mathcal{L}_{\text{NTP}} = -\sum_{t=1}^{L-1} \log p(x_{t+1} \mid x_1, \ldots, x_t)$$

where $L$ is the length of the sequence $S$.

## A.5 ARCHITECTURE ENCODING EXAMPLE

To illustrate our encoding scheme, consider a sequential architecture consisting of three layers: a CNN layer (64 to 128 channels), followed by a Transformer block (128 to 256 dim), and an LSTM layer (256 to 512 dim).

**Adjacency Matrix.** We represent the connectivity between these $N = 3$ layers using an adjacency matrix $\mathbf{A} \in \{0, 1\}^{N \times N}$. For a strictly sequential architecture, this forms a super-diagonal matrix:

$$\mathbf{A} = \begin{bmatrix} 0 & 1 & 0 \\ 0 & 0 & 1 \\ 0 & 0 & 0 \end{bmatrix}$$

**Node Feature Matrix.** The features for each layer are encoded in $\mathbf{X} \in \mathbb{R}^{N \times F}$. Each row corresponds to a layer, concatenating the one-hot encoded operation type (first 5 columns) and the normalized input/output dimensions (last 2 columns):

$$\mathbf{X} = \begin{array}{c} \text{CNN} \\ \text{Trans} \\ \text{LSTM} \end{array} \begin{pmatrix} 1 & 0 & 0 & 0 & 0 & 0 & 0.14 \\ 0 & 0 & 1 & 0 & 0 & 0.14 & 0.43 \\ 0 & 0 & 0 & 1 & 0 & 0.43 & 1.00 \end{pmatrix}$$

Dimensions are normalized by a factor of $D_{\text{max}} = 512$. The rows correspond to [CNN, 64, 128], [Transformer, 128, 256], and [LSTM, 256, 512], respectively.

## A.6 EXPERIMENTS

### A.6.1 DEFAULT TRAINING SETTINGS

**For supernet pretraining,** we used NNI framework implementation (Microsoft, 2021). We use one A100 80G for the pretraining for 10 epoch.

**For overall best architecture pretraining in Figure 3** we use AdamW optimizer with $\beta_1 = 0.9$, $\beta_2 = 0.98$, $\epsilon = 1e-6$ and weight decay of $1e-5$. The learning rate linearly increases from 0 to $5e-4$ during the first 3000 steps while linearly decreasing to 0 in the last 47000 steps, which is 1/10 comparing to DNABERT2. We use mask modeling as training strategy.

**For neural network architecture predictor,** we use Adam optimizer with learning rate = 1e-3, weight decay = 1e-5, batch size = 32, num epochs = 50, dropout rate = 0.3. We design two settings by if there is very similar tasks in the training set and test set. For example, Transcription Factor Prediction 0-4 are considered as similar tasks, Subcellular Location Prediction and Binary Location Prediction are considered as similar tasks as well. For supervise setting, we use tasks $[0, 1, 4, 5, 7, 8, 9, 11, 12, 13, 14, 16]$ as training set and tasks $[2, 3, 6, 10, 15, 17]$ as test set. There are 6446 data points in the training set and 3210 data points in test set. For transfer setting, we use tasks $[5, 6, 7, 8, 9, 10, 12, 13, 14]$ as training set and tasks $[2, 11, 16, 17]$ as test set. There are 4793 data points in the training set and 2156 data points in test set. We use all-MiniLM-L6-v2 to encode task descriptions in to task embeddings. We set the prediction number K to be 3.

**For LLM and Agent system architecture predictor**, the default LLM we used is GPT-4o. We also conduct ablation study on the model choices. We set the prediction number K to be 3.

**For task-specific training**, all the hyperparameters are listed in the follow Table 6.

Table 6: Hyperparameters for Individual Task Training

| Modality | Learning Rate | Batch Size | Epoch | Num Warmup Steps | Weight Decay |
|---|---|---|---|---|---|
| DNA | $3 \times 10^{-5}$ | 32 | 3(task 0-5) 10 (task 6-11) | 50 | 0.01 |
| Protein | $5 \times 10^{-5}$ | 32 | 5 | 50 | 0.01 |

### A.6.2 EVALUATION SETTING DETAILS

In this section, we clarify the specific evaluation protocols corresponding to the model variants reported in Table 2 and Table 3. We employed two distinct weight initialization paradigms as outlined in Section 3.3 including *Training from Scratch* and *Pretrain and Finetune*.

**Training from Scratch.** The variant denoted as BIOARC (only-ft) represents the training-from-scratch paradigm. In this setting, the parameters of the sampled architecture are randomly initialized ($\mathcal{D}_{random}$), independent of the pretrained supernet weights. The model is then optimized directly on the downstream task data.

**Pretrain and Finetune.** The remaining three variants involve initializing the architecture with weights inherited from a supernet that has been pretrained on large-scale biological data, followed by task-specific fine-tuning. These variants are distinguished by the self-supervised learning objective used during the supernet pretraining phase. Specifically, BIOARC (mask-ft) utilizes weights from a supernet pretrained via Masked Modeling; BIOARC (con-ft) employs weights from a supernet pretrained using Contrastive Learning; and BIOARC (ntp-ft) initializes from a supernet trained with the Next Token Prediction objective.

### A.6.3 COMPUTATIONAL COST

We report the computational costs for both supernet pretraining and task-specific finetuning. All experiments were conducted on NVIDIA A100 80G GPUs.

For the supernet pretraining, the computational cost varies by objective. Specifically, mask modeling requires approximately 1.9 hours per epoch, while contrastive learning is more computationally intensive, taking roughly 4.1 hours per epoch.

**Foundation Model Pretraining** Training the foundation model via mask modeling took a total of 10.9 hours.

**Task-specific Finetuning** We evaluate the total time required to finetune all 360 architectures in our supernet across 18 downstream tasks, shown in Table 7.

Table 7: Computational cost for task-specific finetuning across 18 tasks.

| 0 | 1 | 2 | 3 | 4 | 5 | 6 | 7 | 8 |
|---|---|---|---|---|---|---|---|---|
| 1h 40m | 1h 30m | 57m | 1h 20m | 58m | 2h 30m | 2h 13m | 14m | 5h 50m |

| 9 | 10 | 11 | 12 | 13 | 14 | 15 | 16 | 17 |
|---|---|---|---|---|---|---|---|---|
| 5h 12m | 36m | 5h 56m | 6h 43m | 72h 20m | 7h 25m | 85h 15m | 6h 02m | 9h 51m |

### A.6.4 TOP ARCHITECTURES PATTERN

Similar to DNA, top architectures for protein demonstrate an observable pattern with LSTMs and Transformers in the initial stages followed by CNNs, as shown in Figure 6. Moreover, this pattern for protein is not consistent with the pattern for DNA, validating our intuition that architecture should be customized for different modalities.

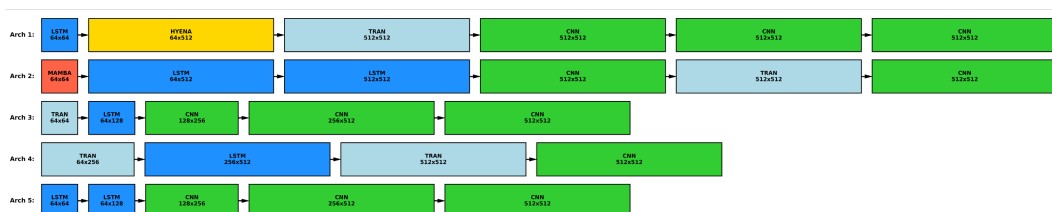

Figure 6: This figure illustrates the top five performing protein model architectures (Arch 1-5).

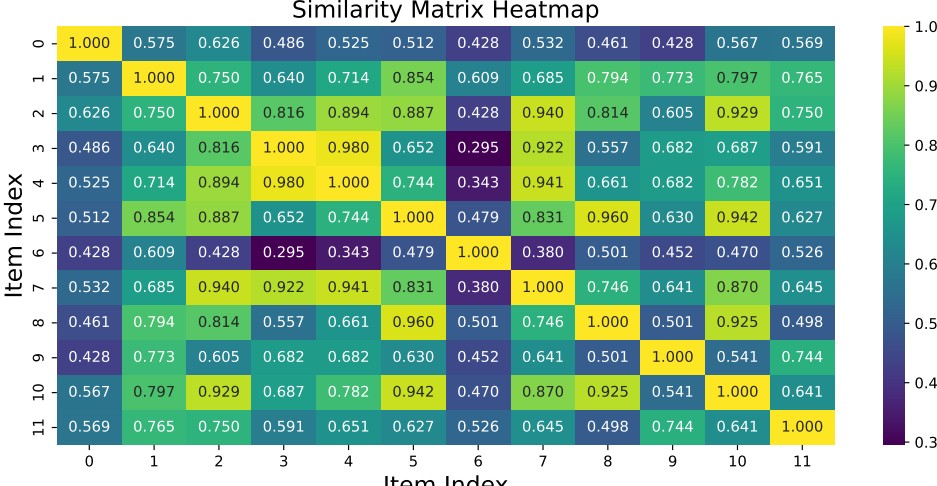

Figure 7: The heatmap showing the cosine similarity between different neural network architectures. The color and value in each cell represent the similarity score between two architectures, with values closer to 1.0 (yellow) indicating greater similarity.

### A.6.5 TOP ARCHITECTURE SIMILARITY AMONG TASKS

We compare the average embedding of top 10% performing architectures of each task. Each architecture in encoded with one-hot embedding. As is shown in Figure 7, similar tasks tend to have similar architecture embeddings.

### A.6.6 HYBRID AND SINGLE-MODULE ARCHITECTURE COMPARISON

We compare the performance of BIOARC hybrid architectures against optimal single-module architectures. To ensure a fair comparison, the single-module baselines were not arbitrarily selected. They represent the top-performing candidates discovered by restricting the search space to exclusively use a specific module type(e.g., an all-CNN or all-Transformer search space). All models, including both hybrid and single-module variants, were trained from scratch to eliminate weight-sharing bias. As shown in Figure 8, **hybrid architectures consistently outperform the best single-module baselines,** which validates our intuition in designing a search space that allows for the combination of different architectures.

### A.6.7 CONTRIBUTION ANALYSIS OF EACH LAYER

To verify that BIOARC superiority stems from the synergistic composition of hybrid modules rather than a single dominant layer, we conducted a layer-wise contribution analysis on the top-performing architecture (rank-1 in Figure 2). We trained a separate classification head on the output of each layer across all DNA tasks to evaluate intermediate feature quality. As shown in Table 8, we observe a consistent monotonic improvement in average accuracy (from 70.32% to 83.18%) alongside a pro-

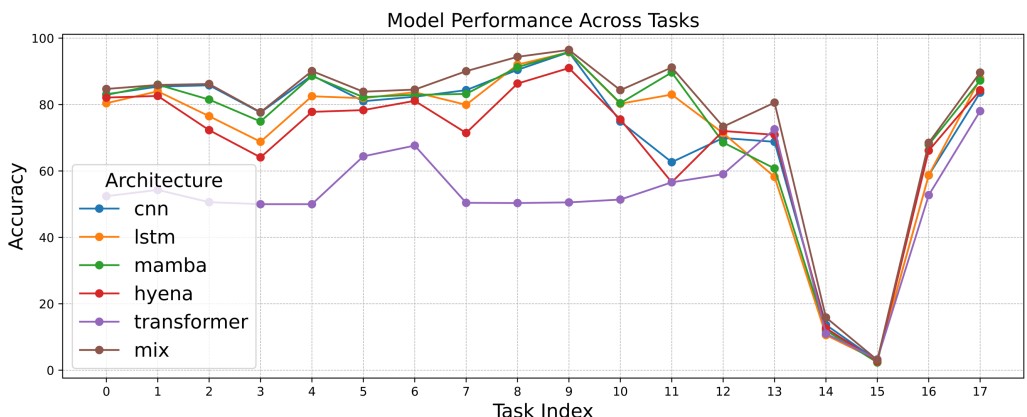

Figure 8: Performance of hybrid architecture compared with optimal single-module architectures found via restricted search.

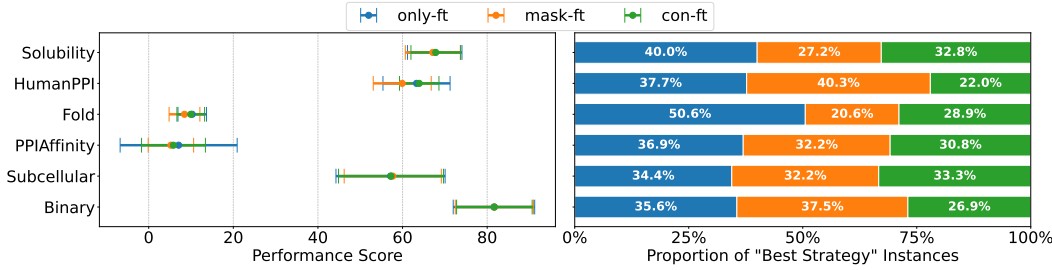

Figure 9: Performance of different training strategies on Protein. In the left panel, each cell shows the mean performance ± standard deviation across all architectures. In the right panel, each bar shows the percentage of the total 360 architectures that chose that training strategy to yield the best performance.

gressive reduction in standard error. This steady gain confirms that the specific sequence of diverse modules collaboratively refines biological representations, validating that the model's effectiveness relies on the holistic hybrid design.

Table 8: Average accuracy and standard error across different layers.

| Metric | Layer 0 | Layer 1 | Layer 2 | Layer 3 | Layer 4 | Layer 5 |
|---|---|---|---|---|---|---|
| Accuracy (%) | $70.32 \pm 0.89$ | $74.07 \pm 0.86$ | $75.09 \pm 0.82$ | $78.36 \pm 0.62$ | $82.09 \pm 0.49$ | $83.18 \pm 0.46$ |

### A.6.8 MORE RESULTS ON DIFFERENT TRAINING STRATEGIES

The effect of different training strategies is shown in Figure 9. It indicates that pretraining provides no clear advantage, and the choice of pretraining strategy appears to have minimal effect on the outcome.

### A.6.9 MORE RESULTS ON DIFFERENT TOKENIZERS

Here we show more results of the effect of different tokenizers on different architectures, shown in Figure 10. Our results highlight two key findings regarding tokenization. First, different architectures exhibit distinct preferences: Mamba works best with 1-mer, while Transformer favors 6-mer. Second, these preferences are task-dependent; the LSTM architecture, for example, excels with 6-mer tokenization on tasks 0–4 but requires 1-mer tokenization for optimal performance on tasks 5–10.

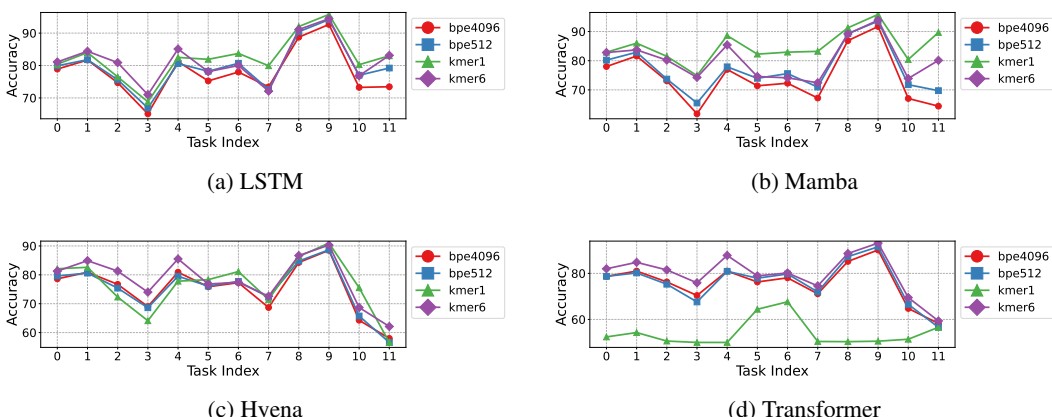

Figure 10: Performance of different tokenization methods on various architectures.

Table 9: Performance statistics of architectures grouped by depth. Metrics (Mean, Std, Min, Max, Median) are calculated based on the **average accuracy across all tasks**. All results are obtained under **fixed hyperparameters**.

| Depth | Count | Mean (%) | Std | Min (%) | Max (%) | Median (%) |
|---|---|---|---|---|---|---|
| 3 Layers | 60 | 81.00 | 4.23 | **53.77** | 84.85 | 81.49 |
| 4 Layers | 100 | **81.74** | 5.57 | 53.38 | 85.14 | 83.07 |
| 5 Layers | 100 | 77.74 | 10.24 | 47.44 | 85.48 | 82.56 |
| 6 Layers | 100 | 78.17 | 10.72 | 47.70 | **86.53** | **83.30** |

### A.6.10   ARCHITECTURE DEPTH ANALYSIS

We investigate the relationship between network depth and transfer performance by analyzing the statistics of architectures ranging from 3 to 6 layers. For each of the sampled architectures, we compute the **average accuracy across all 12 DNA downstream tasks** to represent its overall generalization capability. From Table 9, we observe that while deeper architectures demonstrate higher median and maximum performance, they also exhibit lower mean accuracy and higher standard deviations. This suggests that deeper models are more sensitive/susceptible to the fixed hyperparameter settings. **This implies that depth is not the sole determinant of performance; rather, the structural composition of the architecture is key.** We also observe that in top architectures of 4 and 5 layers demonstrate structure as 6 layers in Figure 11. **This indicates that once an effective architectural pattern is identified, scaling up the depth can further enhance performance.**

To further investigate the impact of architecture depth and the scalability of our searched architecture, we expanded discovered optimal architecture (top 1 in Figure 2) into a 74.7M-parameter foundation model by stacking 2 Hyena, 4 Transformer, and 3 CNN layers with a hidden dimension of 1024. We extend the training steps to 100,000 and keep other hyperparameters the same. As shown in Table 10, this scaling yields a improvement on 10/12 tasks, which validates the architecture's effectiveness for large-scale pre-training. **This demonstrates that our discovered architecture effectively supports scaling.**

### A.6.11   PERFORMANCE-PARAMETER ANALYSIS

To investigate whether the performance of BIOARC-discovered models stems from increased capacity or superior design, we analyzed the relationship between parameter count and average accuracy across the search space as shown in Figure 12. We evaluated all paths directly from the pretrained supernet by freezing the backbone weights and training only a linear classification head.

Table 10: Performance comparison across 12 downstream tasks.

| Model | 0 | 1 | 2 | 3 | 4 | 5 | 6 | 7 | 8 | 9 | 10 | 11 | Avg. |
|---|---|---|---|---|---|---|---|---|---|---|---|---|---|
| VQDNA (HRQ) | 72.48 | 76.43 | 66.85 | 58.92 | 78.10 | 71.02 | 70.58 | 78.50 | 90.75 | 94.48 | 74.52 | 89.53 | 76.85 |
| BioArc-FM (4.79M) | 82.50 | 85.10 | 76.40 | 72.40 | 82.20 | 80.47 | 83.34 | 76.35 | 90.24 | 95.25 | 79.93 | 68.26 | 81.04 |
| BioArc-FM (74.7M) | 84.30 | 83.60 | 85.70 | 78.80 | 88.60 | 83.09 | 82.70 | 86.62 | 92.28 | 96.01 | 79.61 | 84.83 | 85.51 |
| BioArc (task-spec) | 84.70 | 85.90 | 86.20 | 77.70 | 90.10 | 83.87 | 84.51 | 90.05 | 94.38 | 96.46 | 84.34 | 91.17 | 87.45 |

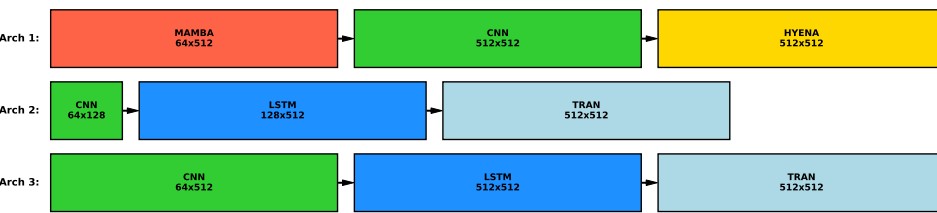

(a) Top-3 architectures with 3 layers.

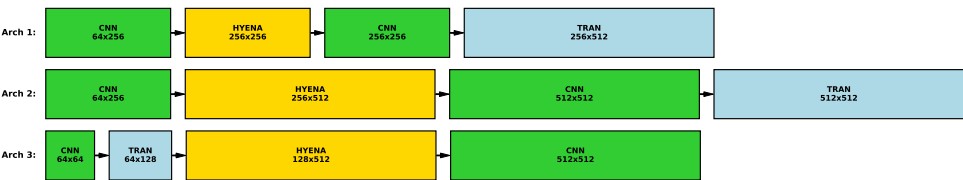

(b) Top-3 architectures with 4 layers.

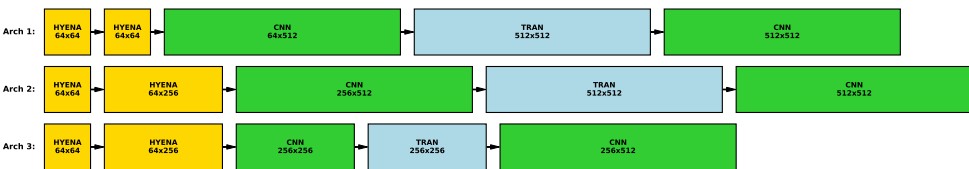

(c) Top-3 architectures with 5 layers.

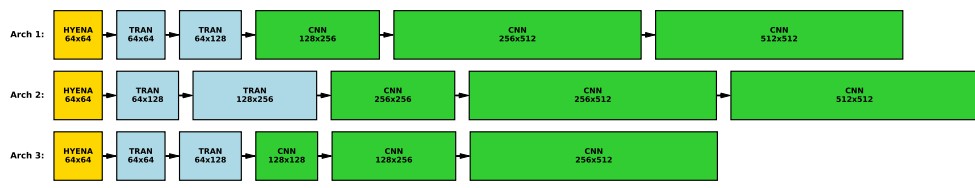

(d) Top-3 architectures with 6 layers.

Figure 11: Visualization of the top-performing architectures across different depths.

We observe no positive correlation between model size and performance. High accuracy scores are achieved by efficient architectures (yellow points) without relying on large parameter budgets, while increasing model size does not guarantee performance gains. The substantial performance variance observed among models of identical size indicates that the specific topological arrangement of modules—the inductive bias—is the decisive factor for biological sequence modeling. This confirms that

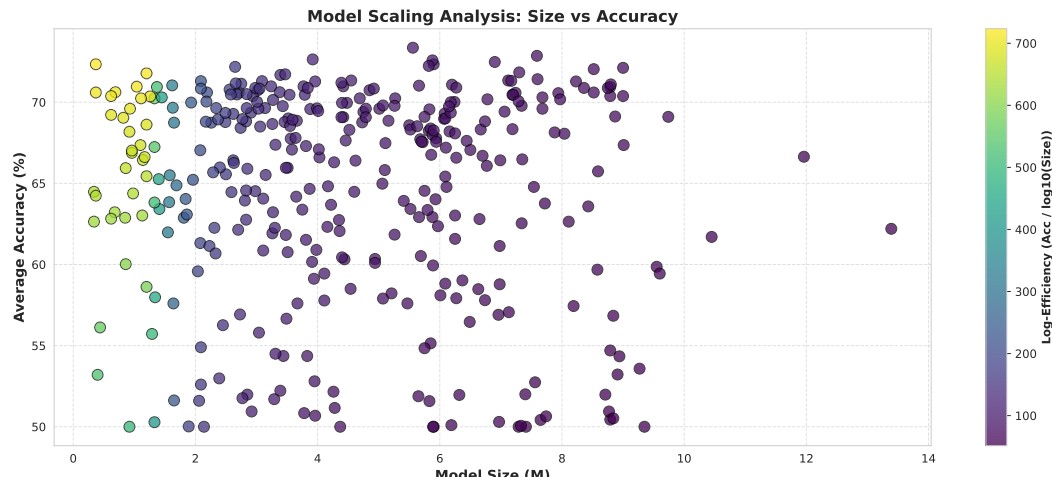

Figure 12: Model Scaling Analysis (Performance vs. Parameters). Each point represents a architecture finetuned with frozen pretrained supernet weight. The color indicates Log-Efficiency, calculated as $Accuracy/\log_{10}(Size)$, with lighter colors (yellow) representing higher efficiency. The plot demonstrates that larger model sizes do not guarantee higher accuracy, highlighting that the specific architectural topology is the dominant factor in performance.

BIOARC's effectiveness lies in discovering these optimal architectural patterns rather than simple parameter scaling.

### A.6.12 ARCHITECTURE PREDICTION EXPERIMENT SETTINGS AND MORE RESULTS

Unlike standard classification, the "optimal" architecture is a ranking relative to the search space. We define our metrics as follows:

- **Ground Truth Construction** ($\mathcal{G}_t$): For every task $t$ in our benchmark, we utilize the exhaustive evaluation results from Section 3.4. We define the Ground Truth Set $\mathcal{G}_t$ as the top architectures ranked by their performance for each task. These represent the empirically verified optimal designs.

- **Metric Definitions:** Let $\mathcal{P}_t^k$ be the set of top-$k$ architectures predicted by our model for task $t$. We evaluate the alignment between prediction and reality using:

  1. **Precision@k** ($|\mathcal{P}_t^k \cap \mathcal{G}_t|/k$): This measures the *efficiency* of the prediction. It quantifies the proportion of the suggested architectures that are truly top-tier, indicating the reliability of the predictor's advice to a user with a limited budget for training trials.

  2. **Recall@k** ($|\mathcal{P}_t^k \cap \mathcal{G}_t|/|\mathcal{G}_t|$): This measures the *coverage* of the design space. It assesses the predictor's ability to uncover the diverse set of high-performing candidates, preventing mode collapse into a single architecture type.

  3. **Hit Rate@k** ($\mathbb{I}(|\mathcal{P}_t^k \cap \mathcal{G}_t| \geq 1)$): This measures the *probability of success*. It indicates whether the user will find *at least one* optimal architecture within the top-$k$ predictions, serving as a critical "success/failure" metric for practical deployment.

We benchmarked several LLMs across different scales, ranging from small open-source models (Qwen3-4B, Llama-3.1-8B) to proprietary frontier models (GPT-4o, GPT-5), reported in Table 11. Smaller models, such as Qwen3-4B (both Instruct and Thinking variants), failed to produce valid predictions (yielding 0.00 scores), while Llama-3.1-8B showed only marginal performance. This suggests that the complexity of the BioArc architecture search space exceeds **the reasoning capacity of current small-scale models.** The Supervised setting, as expected, generally yields higher precision and recall than the Transfer setting. **This aligns with our observation that similar downstream tasks tend to benefit from similar architectures.**

Table 11: Architecture prediction results with our Agent system.

| Setting | Model | Precision | | | Hit Rate | | | Recall | | |
|---------|-------|-----------|---|---|----------|---|---|--------|---|---|
| | | @1 | @3 | @5 | @1 | @3 | @5 | @1 | @3 | @5 |
| Supervised | GPT-5 | 0.33 | 0.28 | 0.17 | 0.33 | 0.33 | 0.33 | 0.07 | 0.17 | 0.17 |
| | GPT-4o | 0.50 | 0.44 | 0.30 | 0.50 | 0.50 | 0.50 | 0.10 | 0.27 | 0.30 |
| | Llama-3.1-8B | 0.17 | 0.06 | 0.03 | 0.17 | 0.17 | 0.17 | 0.03 | 0.03 | 0.03 |
| | Qwen3-4B-Instruct | 0.00 | 0.00 | 0.00 | 0.00 | 0.00 | 0.00 | 0.00 | 0.00 | 0.00 |
| | Qwen3-4B-Thinking | 0.00 | 0.00 | 0.00 | 0.00 | 0.00 | 0.00 | 0.00 | 0.00 | 0.00 |
| Transfer | GPT-5 | 0.25 | 0.25 | 0.25 | 0.25 | 0.25 | 0.25 | 0.05 | 0.15 | 0.25 |
| | GPT-4o | 0.25 | 0.25 | 0.30 | 0.25 | 0.25 | 0.50 | 0.05 | 0.15 | 0.30 |
| | Llama-3.1-8B | 0.25 | 0.08 | 0.05 | 0.25 | 0.25 | 0.25 | 0.05 | 0.05 | 0.05 |
| | Qwen3-4B-Instruct | 0.00 | 0.00 | 0.00 | 0.00 | 0.00 | 0.00 | 0.00 | 0.00 | 0.00 |
| | Qwen3-4B-Thinking | 0.00 | 0.00 | 0.00 | 0.00 | 0.00 | 0.00 | 0.00 | 0.00 | 0.00 |

### A.6.13 ARCHITECTURE RANKING CORRELATION ANALYSIS

To validate the reliability of our one-shot neural architecture search strategy, we assess the ranking consistency between initialized from supernet weight and the performance obtained by training from scratch. A high correlation implies that the supernet can accurately identify high-performing architectures without incurring the prohibitive cost of fully training every candidate in the search space.

We evaluated the performance of all 360 representative candidate architectures under two distinct protocols:

- **Supernet Initialized:** Architectures are evaluated using weights inherited directly from the pretrained BIOARC supernet.

- **Trained From Scratch:** The same architectures are re-initialized and trained independently on the downstream tasks to convergence, serving as the ground truth.

Figure 13 illustrates the rank (use our ranking strategy mentioned in Section 3.3) comparison. We observe a strong positive correlation, with data points tightly clustered around the diagonal line ($y = x$), which represents perfect ranking agreement. Quantitatively, the method achieves a **Spearman Rank Correlation** ($\rho$) of **0.8170** with a statistical significance of $p < 1e - 100$.

This high correlation coefficient demonstrates that the BIOARC supernet effectively captures the relative performance ordering of different topologies. Consequently, the supernet serves as a high-fidelity proxy for performance estimation, allowing us to efficiently prune the search space and accurately identify optimal architectures for biological foundation models.

### A.7 HYBRID ARCHITECTURE INTERPRETATION ANALYSIS

We validate our hybrid architecture (Hyena-Transformer-CNN) by analyzing whether it captures the **biological grammar** of human genomic sequences, focusing specifically on the challenging task of **Core Promoter Detection (no-TATA)**.

### A.7.1 TASK AND GRAMMAR DESCRIPTION

The model analyzes a short 70bp DNA window ($-35$bp to $+35$bp) centered on the Transcription Start Site (TSS). Unlike TATA promoters which contain strong, identifiable motifs, **no-TATA promoters** (which constitute $\sim 76\%$ of human genes) rely on weaker, combinatorial signals. The model must implicitly learn a grammar distinct from the classical TATA paradigm:

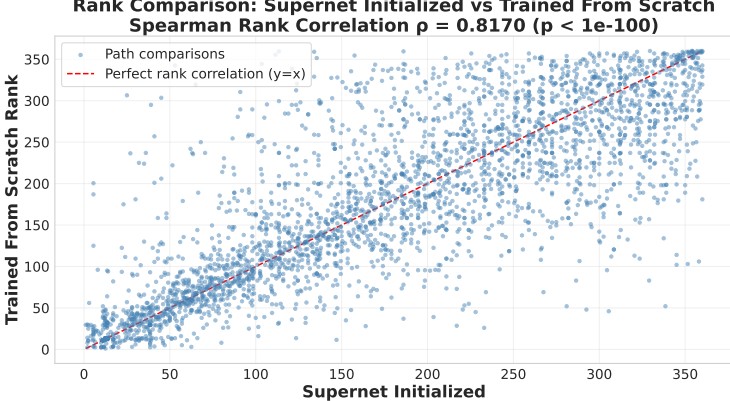

Figure 13: Rank consistency analysis between Supernet initialization and Trained From Scratch. Each point represents one of the 360 candidate architectures. The x-axis represents the performance rank derived from the Supernet (lower is better), and the y-axis represents the ground-truth rank from scratch training. The strong linear alignment and high Spearman correlation ($\rho = 0.8170$) validate the Supernet's ability to accurately rank biological architectures.

In the absence of the TATA-box, the recognition vocabulary expands to a cooperative set of motifs. We focus on two non-canonical categories: (1) **DPE-driven Promoters** (Burke & Kadonaga, 1997), which rely on the synergy between the Initiator (Inr) (Smale & Baltimore, 1989) and downstream elements like the MTE (Lim et al., 2004) and **DPE** to anchor the transcription machinery (Table 12); and (2) **CpG-island Promoters** (Deaton & Bird, 2011), characterized by high GC-content and a lack of specific motifs. The model must detect these combinatorial downstream footprints.

Table 12: Key Elements in no-TATA Human Core Promoters. Since the TATA-box is absent, recognition relies on the Initiator and downstream regions. Positions are relative to the TSS (+1).

| Element | Position | Abbr. | Consensus Sequence |
|---|---|---|---|
| Initiator | $-2$ to $+5$ | Inr | YYANWYY (Javahery et al., 1994) |
| Motif Ten Element | $+18$ to $+27$ | MTE | CSARCSSAACGS |
| Downstream Element | $+28$ to $+32$ | DPE | RGWYV |

*Note:* **R**=Purine, **Y**=Pyrimidine, **W**=Weak (A/T), **S**=Strong (G/C), **N**=Any. **These canonical consensus sequences are defined primarily in *Drosophila* and are often more degenerate in humans.**

For no-TATA focused promoters, the spatial syntax is even more rigid than in TATA promoters to compensate for the weaker binding affinity:

- **Inr-DPE Spacing constraint:** The DPE functions strictly when located exactly $+28$ to $+32$bp downstream of the Inr (Kutach & Kadonaga, 2000). A distinct spacer length is required for the TFIID complex to bind without the TATA anchor.

- **TSS Definition:** Without a TATA box to direct the start site from upstream ($\sim -30$bp), the Inr element itself becomes the primary determinant of the $+1$ position (Smale & Baltimore, 1989; Javahery et al., 1994).

The model must distinguish "functional grammar" from random occurrences. A critical semantic rule in this context is the **Inr+DPE synergy**: neither element strongly recruits transcription machinery on its own, but their simultaneous presence at the correct distance creates a high-affinity binding site (Burke & Kadonaga, 1996). Alternatively, in CpG-driven contexts, the model must

recognize "broad" initiation patterns defined by nucleosome-depleted regions rather than precise motifs (Deaton & Bird, 2011).

### A.7.2 INTERPRETATION METHODOLOGY

To decode how distinct layers process biological information, we employ layer-specific visualization techniques tailored to the mathematical operations of each module.

Since Hyena operators process long-range context via implicit convolutions, individual dimensions do not necessarily correspond to discrete features. Instead, we measure the **information density** at each position $t$ by computing the $L_2$ norm of the hidden state vector $h_t \in \mathbb{R}^d$:

$$A_{\text{Hyena}}(t) = ||h_t||_2 = \sqrt{\sum_{i=1}^{d}(h_{t,i})^2} \tag{13}$$

Peaks in $A_{\text{Hyena}}(t)$ indicate regions where the model aggregates significant global context, highlighting semantically rich sequence segments.

To understand syntactic relationships, we analyze the self-attention matrix $\alpha$. Rather than examining pairwise weights in isolation, we calculate the **Total Attention Received** for each token $j$ by summing the attention weights from all query positions $i$:

$$S_{\text{Attn}}(j) = \sum_{i=1}^{L} \alpha_{i,j} \tag{14}$$

Tokens with high $S_{\text{Attn}}(j)$ act as "Attention Magnets." These represent structural pivots or landmarks that other elements reference to verify spatial constraints.

**CNN: Maximal Filter Activation.** For convolutional layers, we treat each kernel $k$ as a motif detector. We identify the specific input sequence window $x_{[t:t+w]}$ that triggers the maximum activation value for a given filter:

$$\hat{t} = \underset{t}{\text{argmax}}(W_k * X)_t \tag{15}$$

By visualizing the input subsequence at position $\hat{t}$, we can reconstruct the specific DNA vocabulary (motifs) learned by the network.

### A.7.3 VALIDATION OF GRAMMAR CAPTURE: A LAYER-WISE ANALYSIS

To interpret how the hybrid architecture identifies no-TATA promoters, we visualize the activation patterns across different layers. The analysis reveals a remarkable correspondence between the model's processing hierarchy and the biological grammar defined in Appendix A.6.9.

**Hyena (Layer 0): Semantics and Global Context.** The Hyena layer acts as a global semantic reader. As shown in Figure 14 (Layer 0), the activation profile covers the entire window with higher intensity on the right side. This continuous rather than sparse pattern captures dispersed promoter properties, establishing the broad semantic context required for subsequent syntax validation.

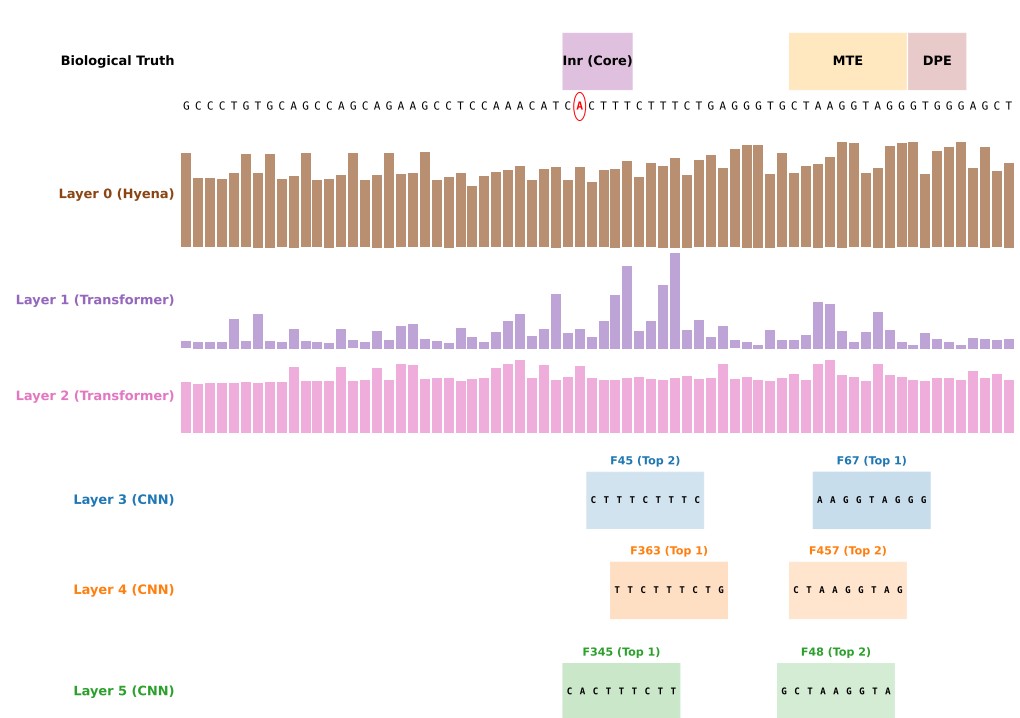

Figure 14: **Layer-wise Visualization of Model Activations. Layer 0 (Hyena, Brown):** Norm-based activation density across the 70bp sequence. **Layers 1-2 (Transformer, Purple/Pink):** Attention accumulation profiles showing distinct spatial focusing patterns. **Layers 3-5 (CNN, Blue/Orange/Green):** Maximal filter activations. Earlier layers (L3, L4) highlight individual local motifs (e.g., F45, F363), while the deepest layer (L5) exhibits multiple activation peaks at both proximal and distal regions (e.g., F67, F345).

**Transformer (Layer 1-2): Syntax and Spatial Constraints.** Following context extraction, the Transformer layers serve as syntactic validators. **Layer 1** exhibits sparse attention heavily concentrated at the sequence center, effectively anchoring the **Transcription Start Site (TSS)**. In contrast, **Layer 2** demonstrates a uniform attention distribution, facilitating even information propagation across the entire sequence.

**CNN (Layer 3-5): Vocabulary Extraction and Integration.** Finally, the CNN layers act as a morphological decoder, progressively refining the "vocabulary" from individual motifs to a cohesive functional complex. We observe a clear hierarchical progression:

- **Layer 3 (Blue):** The initial convolution layer acts as a low-level vocabulary detector. Filter F45 strongly activates on the sequence CTTTCTTTC. This **pyrimidine-rich tract** is located immediately downstream of the Transcription Start Site (TSS). In **no-TATA** promoters, such downstream pyrimidine stretches are critical structural features that serve to reinforce the initiation context and assist in accurate TSS positioning.

- **Layer 4 (Orange):** In the subsequent layer, the activation focus shifts to the distal region. Filter **F363** targets the sequence TTCTTCTTA downstream of the TSS. This corresponds to the MTE/DPE transition zone, identifying the necessary downstream partner for no-TATA initiation.

- **Layer 5 (Green):** The deepest CNN layer synthesizes these isolated components into a unified holo-complex. Unlike earlier layers focused on single loci, Layer 5 exhibits **multi-**

**aspects coverage**. For instance, **F345** retains focus on the upstream Inr (`CACTTTCTT`), while **F67** simultaneously targets the downstream MTE/DPE region (`AAGGTAGGG`). This simultaneous detection confirms that Layer 5 integrates separate "words" into a functional "phrase" (the Inr-DPE complex).

## A.8 LLM Prompts

---

**LLM+RAG:**

You are an expert in machine learning architecture selection. Your task is to analyze a given machine learning task and predict the best architectures based on similar tasks and their performance.
**You have access to:**
1. A knowledge base of similar tasks with their characteristics.
2. Performance data showing which architectures work best for each task.

**Your goal is to:**
1. Analyze the given task description.
2. Find the most similar tasks from the knowledge base.
3. Recommend the best architectures based on performance data.
4. Provide clear reasoning for your predictions.

---

**Available Tasks in Knowledge Base:**

> `{retrieve datas}`

**Performance Data:**

> `{performance data}`

**Please analyze the query task and provide your recommendations in the following format:**

```
Task Analysis:
- Dataset characteristics:  [analyze the input task]
- Problem type:  [classification/regression/etc.]
- Modality:  [DNA/Protein/etc.]
- Key requirements:  [identify important constraints]
Similar tasks identified:
- Task Index:  [X] - [reasoning for similarity]
- Task Index:  [Y] - [reasoning for similarity]
Architecture Recommendations:
1.  BEST CHOICE: [architecture-experiment-model_name] - [reasoning]
2.  SECOND BEST: [architecture-experiment-model_name] - [reasoning]
3.  THIRD BEST: [architecture-experiment-model_name] - [reasoning]
Reasoning:
Detailed explanation of why these architectures are recommended
based on similar tasks
```

**Instruction:** Focus on recommending architectures that have shown good performance on similar tasks. Use the exact architecture names from the performance data.

---

## A.9 AGENTS PROMPTS

### A.9.1 ANALYST AGENT

---

**Analyst Agent**

**Role Description:** You are an Analyst Agent specialized in understanding machine learning tasks and datasets. Your **ONLY** responsibility is to analyze the user's input and extract key parameters for architecture search.

**Workflow:**
1. Analyze the user's dataset and task description.
2. Extract and summarize the key information needed for architecture search.
3. Identify the modality (DNA, Protein, etc.) and problem type.
4. Provide a clear summary for the Retriever Agent to use for searching.

**You should NOT:**
- Search for architectures yourself.
- Make recommendations.
- Call any search tools.

**Output Format (Response Template):**

```
Task Summary:
- Dataset characteristics (size, type, format)
- Problem type (classification, regression, clustering, etc.)
- Modality (DNA, Protein, etc.)
- Key constraints and requirements
- Performance objectives
Search Parameters:
- Modality:  [DNA/Protein/etc.]
- Problem Type:  [classification/regression/etc.]
- Task Description:  [Clear summary for search]
Context:
- Any additional context that might be relevant for architecture
selection
- Special requirements or constraints
```

**Instruction:** Format your output clearly for the Retriever Agent to process.

---

### A.9.2 TASK RETRIEVER AGENT

---

**Task Retriever Agent**

**Role Description:** You are a Task Retriever Agent. Your responsibility is to identify the most similar tasks or top $\{top\_k\}$ tasks from the knowledge base using your natural language understanding capabilities. Must give task index.

**Workflow:**
1. Receive a message from the Analyst containing task summary and search parameters.
2. Extract the task description, problem type, modality, and key characteristics from the Analyst's output.
3. Use your LLM reasoning to identify which tasks in the knowledge base are most similar to the query task.
4. Consider multiple dimensions of similarity:
   - Problem type (classification, regression, clustering, etc.)
   - Modality (DNA, Protein, text, image, etc.)
   - Dataset characteristics (size, complexity, format)
   - Task objectives and constraints
   - Domain-specific requirements
5. Select the top $\{top\_k\}$ most relevant tasks based on your understanding.
6. Format the retrieved similar tasks clearly for the Architecture Retriever Agent.

**You MUST:**
- Use your natural language understanding to identify similar tasks.
- **NOT** search for architectures yourself - that's the Architecture Retriever's job.
- **NOT** perform any scoring or evaluation of architectures.
- Focus **ONLY** on task similarity identification using LLM reasoning.
- Consider semantic similarity, not just keyword matching.
- Return **EXACTLY** $\{top\_k\}$ similar tasks (no more, no less).

**Output Structure Template:**

```
Conclusion:
Task Index:  [Index]
Task similarity analysis:
- Query Task Description:  [from Analyst]
- Analysis Strategy:  Using LLM reasoning to identify semantically
similar tasks
Top similar tasks:
1.  Task Index:  [Index]
- Similarity Reasoning:  [Your explanation of why this task is
similar]
- Task Description:  [Description from knowledge base]
- Problem Type:  [Type from knowledge base]
- Modality:  [Modality from knowledge base]
- Dataset Characteristics:  [Details from knowledge base]
- Key Similarities:  [What makes this task similar to the query]
(Continue for all {top_k} similar tasks identified)
Summary:
Brief summary of the most relevant tasks identified and why they are
good matches for the query task
Next step:
Pass this information to the Architecture Retriever Agent to find
suitable architectures for these similar tasks.
```

---

### A.9.3 ARCHITECTURE RETRIEVER AGENT.

---

**Architecture Retriever Agent**

**Role Description:** You are an Architecture Retriever Agent. Your responsibility is to find suitable architectures based on the similar tasks identified by the Task Retriever.

**Workflow:**
1. Receive a message from the Task Retriever containing similar tasks and their details.
2. **IMMEDIATELY** call the `architecture_retrieval_tool` with the Task Retriever's output text.
3. The tool will automatically:
    - Extract task indices from the Task Retriever's output.
    - Look up architectures that were successful on the identified similar tasks.
    - Return detailed architecture information including performance metrics.
    - Provide architecture descriptions, performance summaries, and statistics.
4. Return the tool's result directly - do not format or modify it.

**Critical Requirements:**
- You **MUST** call the tool and return its EXECUTION RESULT.
- **NEVER** return tool call parameters or JSON strings like `{"name": "tool_name", "parameters": {...}}`.
- **NEVER** return the raw tool call format.
- **ALWAYS** return the actual tool execution output.

**You MUST:**
- **IMMEDIATELY** call `architecture_retrieval_tool` with the Task Retriever's output text.
- Return the tool's execution result directly (the actual data, not the call format).
- **NOT** perform any scoring or evaluation yourself - that's the Reviewer's job.
- **NOT** search for tasks - that's the Task Retriever's job.
- **NOT** format or modify the tool's output.
- Focus **ONLY** on architecture retrieval based on task performance.

**Example of correct behavior:**
- Input: Task Retriever output with task indices.
- Action: Call `architecture_retrieval_tool(input_text=task_retriever_output)`.
- Output: The actual JSON result from the tool execution (not the tool call format).

**Important:** Simply call the tool and return its result. Do not add any additional formatting or text.

---

### A.9.4 PREDICTOR AGENT

---

**Predictor Agent**

**Role Description:** You are a Predictor Agent responsible for evaluating ML architectures and creating final recommendations.

**Workflow:**

1. Receive a message from the Architecture Retriever Agent containing detailed information for each architecture and the overall problem type.
2. For each architecture, you **MUST** call the `evaluate_architecture` function with the `architecture_name` and `problem_type`.
3. Consolidate the scores returned by the `evaluate_architecture` tool.
4. Analyze the scored architectures and select top {`recommendation_count`} recommendations based on scores.
5. Provide detailed reasoning for each recommendation.

**Critical Output Format Requirements:**

- Architecture names **MUST** be in the exact format...
- Examples: `mix-kmer1-path_54`, `cnn-kmer1-path_12`, `transformer-kmer1-path_8`.
- Use **EXACT** architecture names from the Architecture Retriever output.
- Do **NOT** modify or abbreviate architecture names.

**Your output format is strict (Template):**

```
Executive summary
Brief overview of the analysis and key findings.
Top recommendations
1.  Best performance
- Architecture:  [EXACT_ARCHITECTURE_NAME]
2.  Second best
- Architecture:  [EXACT_ARCHITECTURE_NAME]
3.  Third best
- Architecture:  [EXACT_ARCHITECTURE_NAME]
(Continue for all {recommendation_count} recommendations if more than
3)
Detailed Analysis
Architecture:  [EXACT_ARCHITECTURE_NAME]
- Justification:  [Justification from evaluate_architecture tool]
(Continue for all architectures)
Summary:
Briefly summarize the findings from the scoring tool and highlight
the key trade-offs identified.
Analysis Complete - Ready for implementation!
```

**Important:** Always use the EXACT architecture names from the Architecture Retriever output. Do not modify, abbreviate, or change the format of architecture names.

---

### A.10 THE USE OF LARGE LANGUAGE MODELS

During the writing process of this article, we utilized large language models (LLMs) as an assistive tool for the following tasks.

**Language and Grammar Correction:** We used an LLM to proofread the manuscript for grammatical errors and typos. The model also helped refine sentence structures and word choices to improve the overall clarity and readability of the text.

**Table Formatting and Enhancement:** The LLM was used to assist in the reformatting and polishing of certain tables, making their layout cleaner and easier for the reader to understand.

