# OpenReview forum: "BioArc: Discovering Optimal Neural Architectures for Biological Foundation Models"
_ICLR.cc/2026/Conference — Submitted to ICLR 2026_

### Official Review · Reviewer_Rr5N · 2025-10-31

**Soundness:** 2
**Presentation:** 2
**Contribution:** 2
**Rating:** 2
**Confidence:** 2

**Summary:**

The paper introduces BIOARC, a Neural Architecture Search (NAS) framework intended to discover optimal, specialized architectures for biological foundation models (BFMs). The authors rightly criticize the common practice of directly repurposing Transformer architectures from NLP for biological data. The ambition is to provide a "principled, automated architecture discovery" methodology.

To this end, they conduct an extensive NAS over a hybrid design space (combining CNN, Transformer, and Hyena/Mamba blocks) using a one-shot, weight-sharing supernet. This search is evaluated on multiple DNA and protein tasks, considering various tokenization and pretraining strategies.

The results are impressive: the discovered architectures achieve state-of-the-art performance while being considerably smaller than baselines like. The paper also presents valuable findings on the interplay between architecture, tokenization, and pretraining.

**Strengths:**

- Clear presentation: The paper is well-structured and clearly written. The figures are straightforward


- Impressive Empirical Performance: The primary strength is the reported results. The discovered BIOARC architectures (Tables 1 & 2) consistently and significantly outperform much larger baselines (e.g., DNABERT-2, ESM-1b) across DNA and protein tasks. Achieving this with models that are an order of magnitude smaller (e.g., <5M vs >100M parameters) is a highly significant practical contribution.


- Comprehensive Evaluations: The authors are to be commended for the breadth of their study, evaluating their framework across:
    - two distinct biological modalities (DNA and protein).
    - multiple downstream tasks (e.g., TFP, SSP, Fold prediction).
    - The interplay of three key axes: architecture, tokenization, and self-supervised training strategy (MM vs. CL vs. from-scratch).


- Dependence of Tokenizer of architecture: The finding that the optimal tokenizer is highly architecture-dependent (Sec 4.4, Fig 5) is a valuable actionable insight that can be explored further.

**Weaknesses:**

- The paper's entire premise is to move away from "intuition-driven design" However, after discovering a "Hyena→Transformer→CNN" pattern (Fig 2), the authors' only explanation is a classic, post-hoc intuitive rationalization: "Hyena block to first captures long-range dependencies, with Transformer blocks in the middle to model complex contextual relationships and CNN blocks at the end to extract critical features" (Lines 384-386). This argument fundamentally undermines the paper's central claim. They have not delivered a "principled methodology" for discovery, but rather an empirical search which is not as principled as the authors claim.

- Unaddressed Weight-Sharing Bias: The entire NAS relies on a "Single Path One-Shot" (Line 229) supernet with weight sharing, which can introduce significant biases. The discovery of mixed-block architectures may be a direct artifact of the weight-sharing scheme itself, not a reflection of the task's intrinsic needs. For example, do mixed architectures simply converge more reliably, or access a more effective parameter space within the supernet, than pure-block architectures? Without extensive ablations training individual architectures (pure and mixed) from scratch outside the supernet, or validation on toy problems with known optima, it is impossible to know if the "Hyena→Transformer→CNN" pattern is a genuine discovery or just an artifact of the search methodology.


- The introduction of a GPT-4o-based "Agent" (Sec 3.4, 4.5) is also problematic as it might rediscover existing ML literature without acknowledgment. The agent's "superior performance" (Table 3) is the most baffling part. It dramatically outperforms a "Neural Network Predictor" and even a standard "LLM+RAG" (which also uses GPT-4o). The only explanation given is that "specific roles and task definitions... unlocks generalization potential" (Line 480). This is far too vague. What are these roles? What are these definitions? This "agentic framework" seems to be the entire source of the performance lift, yet it is completely sufficiently explained, and potentially unreproducible.

- The metrics in Table 3 ("Hit Rate," "Precision") are poorly defined. What constitutes the "ground truth" set of top architectures for these calculations? This section feels more like a gimmick than a serious methodological proposal.

- Mamba vs. Hyena: Line 216 explicitly states the supernet module types are "e.g., CNN, Mamba". However, Figure 2 and the entire subsequent analysis (e.g., Line 384) refer only"HYENA" blocks. Mamba blocks are not mentioned or discussed again after figure 1.

- Missing Statistical Rigor: For a paper that relies on comparing performance across many architectures, there are no confidence intervals, standard deviations on most tasks (Fig 4 is an exception), or statistical significance tests. The results in Tables 1 & 2 are presented as single-run point estimates. We have no way of knowing if the "top" architectures are statistically superior to any others, or just a result of random seed variation.

- The claim to be the first to move "beyond intuition" is overly broad. The paper fails to properly situate itself against the large body of work on hybrid architectures (e.g., CNN-Transformer or SSM-Transformer models), which are already common.

**Questions:**

1. Addressing the "Principled Discovery" Contradiction: How do you reconcile your central claim of moving beyond "intuition-driven design" with the fact that your primary "discovery" (the Hyena→Transformer→CNN pattern) is justified only by a post-hoc, intuitive argument (Lines 384-386)?

2. Proving the Discovery is Not a NAS Artifact: How can you prove that your discovered "Hyena→Transformer→CNN" pattern is a genuine property of the task and not an artifact of the one-shot, weight-sharing NAS? Would you be willing to provide results from training a selection of pure-block (e.g., all-CNN, all-Hyena) and mixed-block architectures completely from scratch (i.e., new random initializations, not from the supernet) to show the mixed-block models still win?

3. Please provide more details and intuition surrounding the "Agent" Mechanism in the main text.

4. Clarifying Prediction Metrics: Can you please provide a precise definition for the "Hit Rate @k" and "Precision @k" metrics in Table 3?

5.Can you please clarify the inconsistency between Line 216 (which specifies "Mamba") and Figure 2/Section 4.2 (which specifies "Hyena")? Why is Mamba not mentioned after figure 1?

6. Justifying the LLM's Bias: How do you address the fundamental concern that using an LLM (trained on existing ML literature) to predict "good" architectures will inherently bias your search towards existing design patterns, potentially defeating the purpose of discovering novel ones?

7. Biological Insight: The abstract claims current models "struggle to capture... complex underlying 'grammars' inherent to biological data." Could you discuss how this work addresses the grammar of biological data?

---

> ### Author Response · Authors · 2025-11-26
>
> We thank Reviewer Rr5N for the constructive feedback and valuable insights. We are encouraged that the reviewer acknowledges we have impressive empirical performance, comprehensive evaluation and provide valuable insights. We acknowledge the concerns regarding the writing clarity and have carefully polished the presentation to ensure our ideas are conveyed more effectively.
>
> Here we address your concerns.
> ***
> ***
> ### **W1\Q1 post-hoc intuitive rationalization**
> We thank the reviewer for this sharp observation, which allows us to clarify the core definition of our "principled methodology." We respectfully argue that there is no contradiction between our automated discovery process and the post-hoc physical interpretation of its results.
>
> **Principled Methodology.**
> Our use of the term "principled" refers strictly to the BioArc framework itself rather than the specific architectural outputs. As the optimal architecture is highly dependent on modality and task, we do not propose the Hyena $\to$ Transformer $\to$ CNN sequence as a principled, universally optimal architecture. For example, optimal architecture for Protein in Figure 6 is different from Hyena->Transformer->CNN.
>
> **Prior Design vs Posterior Interpretation.**
> The "Hyena then Transformer then CNN" pattern observed in Figure 2 was not a prior assumption encoded into our search space. Our search space included a diverse set of modules like CNN, LSTM, Transformer, Mamba, and Hyena. It allowed for their mutual combination without enforcing a specific order. The algorithm automatically converged to this specific topology. The explanation referenced (long-range dependencies...) is a post-hoc analysis intended to interpret why the algorithm chose this structure but not how we designed it. The fact that the machine-discovered architecture aligns with biological reasoning serves as a validation that the search algorithm is capturing genuine physical signals rather than fitting to noise.
>
> **Interpretation Analysis.**
> We added a new section A.7 to validate that our automatically discovered architecture is indeed aligned with underlying biological grammar in a case study.
> ***
> ***
> ### **W2\Q2 Proving the Discovery is Not a NAS Artifact**
> We appreciate the reviewer's scrutiny regarding the potential bias introduced by weight sharing in the supernet. We fully agree that validating architectures outside the supernet is essential to confirm they are not artifacts.
> We have already performed the exact ablation study. It is one of our findings that mixed architectures outperform architectures with only one module type. In Appendix A.6.5, we provided the detailed settings for the comparison of our discovered "Hybrid" architectures against "Single-Module" architectures, which are the top-performing architectures discovered by restricting our search space to exclusively use a specific module type, such as an all-CNN search space or an all-Transformer search space. We then trained these optimal pure-block candidates from scratch alongside the mixed-block candidates to ensure a fair comparison. The results are presented in Figure 8.
> ***
> ***
> ### **W3\Q3 BioArc agent mechanism**
> We thank the reviewer for highlighting the need for greater clarity regarding the BioArc Agent. We admit that the original description primarily focused on the workflow rather than the underlying intuition and evaluation rigor. We address the concerns as follows:
>
> **Motivation.** BioArc agent is designed to solve the challenge of predicting architectures for novel tasks. It is built on our core experimental observation that optimal architectures for functionally similar tasks share significant structural similarities (Section 4.2). The Agent is not a "black box" magic; it is specifically engineered to exploit this cross-task similarity. We have now detailed our design in Section 3.5.
>
> **Mechanism.** BioArc agent is required to make prediction in our searched space. Based on our observation that similar tasks have similar optimal architecture, we want to find the similar tasks that our architectures have been trained on so we are know the (task, architecture, performance) tuple.
>
> **Interpretation.** The effectiveness of BioArc agent framework comes from: 1. Semantic understanding ability to identify more similar tasks comparing to embedding based RAG with input of multiple aspects. 2. Task decomposition makes it easier for models to reason.
> ***
> ***

---

> > ### Author Response · Authors · 2025-11-26
> >
> > ***
> > ***
> > ### **W4\Q4 Metrics and ground truth explanation**
> > To resolve this ambiguity, our "Ground Truth" refers to the ranking of top-performing architectures on our target task, obtained through empirical verification during the search. The metric definitions are as follows: Hit Rate@N measures if our N prediction contain at least one of the "Ground Truth Top-K" architectures. Precision@N measures the proportion of our N prediction that are correct (i.e., truly belong to the "Ground Truth Top-K" set), thus gauging the accuracy of our prediction. Finally, Recall@K measures the proportion of the K "Ground Truth" top architectures that we successfully identified, gauging the completeness or coverage of our recommendations.
> > ***
> > ***
> > ### **W5\Q5 Mamba vs. Hyena**
> > We thank the reviewer for this keen observation and apologize for any confusion. Line 216 correctly defines our broad search space, which includes CNN, Mamba, and Hyena as potential candidate modules. In contrast, Figure 2 (and Section 4.2) depicts the optimal search result specifically for DNA tasks. Our NAS algorithm determined that Hyena was the optimal choice for this specific modality and constraint, which is why Hyena appears in Figure 2 instead of Mamba.
> > ***
> > ***
> > ### **W6 Missing Statistical Rigor**
> > We thank the reviewer for this critical point regarding statistical validation.
> >
> > **Context for Single-Run Protocol.**
> > Our initial presentation of single-run point
> >  estimates aligns with the standard evaluation protocols of the large-scale biological foundation models we benchmark against, such as DNABERT2 1and VQDNA2. This approach is widely adopted in the field primarily due to the prohibitive computational cost. As detailed in Appendix A.6.3 (Table 7) 3, training a single architecture on computationally intensive tasks (e.g., Task 15) can take over 85 GPU hours4. Conducting multi-seed runs for the entire search space (360 architectures) across all tasks was computationally infeasible.
> >
> > **New Statistical Validation.**
> > We share the reviewer's concern and agree that demonstrating robustness against random seed variation is essential. To validate our findings, we selected the top-1 architectures identified for each DNA tasks (Task 0-11) and trained each of them from scratch 5 times using different random seeds.
> > The results (detailed in the table below) confirm that our architectures exhibit stable performance. The original single-run results fall consistently within the range of the multi-run averages, and the standard deviations are low (average std $\approx 1.05$), confirming that the superior performance is not an artifact of random variation.
> >
> > | 0 | 1 | 2 | 3 | 4 | 5 | 6 | 7 | 8 | 9 | 10 | 11 |
> > | :--: | :--: | :--: | :--: | :--: | :--: | :--: | :--: | :--: | :--: | :--: | :--: |
> > | 84.70 | 85.90 | 86.20| 77.70 | 90.10 | 94.38 | 96.46| 84.34 | 83.87 | 84.51 | 90.05 | 91.17 |
> > | 84.37 (1.14) | 86.87 (0.76) | 86.55 (1.45) | 77.85 (0.85) | 90.30 (0.78) | 94.03 (1.23) | 97.22 (1.25) | 83.94 (1.06) | 83.41 (0.74) | 84.27 (1.23) | 89.43 (0.98) | 90.71 (0.90) |
> > ***
> > ***
> > ### **W7 Comparison with other works on hybrid architectures**
> > We thank the reviewer for highlighting the rich literature on hybrid architectures. We acknowledge that combining blocks (e.g., CNN-Transformer) is an established concept. However, we respectfully clarify that our contribution of moving "beyond intuition" refers to the automated architecture discovery process, not the existence of hybrid models.
> >
> > **Manual Hybridization vs. Automated Discovery**
> > Existing hybrid models typically rely on fixed, manually designed patterns based on human priors (e.g., intuitively placing CNNs first for feature extraction). In contrast, BioArc treats modules (CNN, Mamba, Hyena, Transformer, LSTM) as atomic primitives within a vast, unbiased search space. We do not presume which combination is best; rather, our framework automatically discovers the optimal permutation, depth, and width driven strictly by data signals.
> >
> > **Granularity of Optimization**
> > BioArc optimizes the architecture at a much finer granularity. It determines not just which blocks to use, but their specific ordering and interaction. A compelling example is the discovered Hyena -> Transformer -> CNN sequence for DNA (Figure 2). This topology defies the conventional "CNN-first" intuition in some hybrid architecture works.
> >
> > **Action in Revision**
> > We have expanded the Related Work section to explicitly discuss existing hybrid architectures. We now clearly position BioArc as a framework that systematizes the design of such hybrids, providing a rigorous methodology to navigate the complex design space beyond manual heuristics.
> > ***
> > ***

---

> > > ### Author Response · Authors · 2025-11-26
> > >
> > > ***
> > > ***
> > > ### **Q6 Justifying the LLM's Bias**
> > > We address this concern in two fold.
> > >
> > > **Limited prediction space.** We only allow predictions in our search space for which we have ground truth (task, architecture, performance) tuples. This allows the LLM to reason based on the similarity of task and architecture features.
> > >
> > > **Task decomposition.** Agents exhibit reduced hallucination rates due to the simplicity of step-wise instructions. In contrast, LLM + RAG systems are more prone to confusion stemming from conflicts between retrieved context and pretrained parametric knowledge.
> > > ***
> > > ***
> > > ### **Q7 Biological Insights**
> > > We thank the reviewer for this profound question. We would like to clarify that BioArc does not attempt to "address" biological grammar by encoding explicit linguistic rules by human design, which is limited by incomplete understanding of the physicochemical "syntax" of biology. Instead, BioArc addresses biological grammar by treating it as an unknown data distribution to be fitted. We have added a section in Appendix A.7 Hybrid Architecture Interpretation Analysis.
> > >
> > > In A.7, we tried to visualize the activation, attention accumulation and maximal filter activations of Hyena, Transformer and CNN respectively. On the task of Core Promoter Detection (no-TATA), in which biological grammar is of dominant importance for the prediction, We observed a clear correspondence between the model layers and the hierarchical biological grammar on this specific task as shown in Figure 14. A detailed grammar description, derived from existing biological knowledge relevant to this task, can be found in Appendix A.7.
> > >
> > > **Semantics**: Hyena layers capture global context and combinatorial rules. It can be seen in the figure that the activation distribution for Hyena (Layer 0) covers the entire sequence.
> > >
> > > **Syntax**: Transformer layers enforce strict spatial and positional constraints. The first Transformer (Layer 1) exhibits sparse attention heavily concentrated at the sequence center, effectively anchoring the Transcription Start Site (TSS). The second Transformer (Layer 2) demonstrates a uniform attention distribution, facilitating even information propagation
> > > across the entire sequence.
> > >
> > > **Vocabulary**: CNN layers extract specific local motifs (e.g., Inr, DPE). CNN layers constantly focuses on important local areas such as Inr and MTE, which is core feature for the task.
> > >
> > > We agree that further interpretation and the development of a comprehensive methodology for understanding discovered architectures would be extremely valuable. Specifically, utilizing the optimal architecture to interpret unknown sequence features is a crucial next step, which we designate as important future work.
> > > ***
> > > ***

---

### Official Review · Reviewer_KRGZ · 2025-11-07

**Soundness:** 3
**Presentation:** 3
**Contribution:** 3
**Rating:** 6
**Confidence:** 3

**Summary:**

This paper introduces **BIOARC**, a framework for **automated architecture discovery** in biological foundation models. Unlike prior work that repurposes architectures from general AI domains (like NLP and CV), BIOARC is designed specifically for the **unique physicochemical and structural characteristics** of biological data. Existing foundation models in biology often underperform because they reuse architectures meant for text or images without adapting to biological data’s structure. Furthermore, they fail to capture **long-range dependencies**, **sparse signals**, and **complex biological “grammars.”**

BIOARC addresses these limitations through a **systematic, data-driven approach** to model design. BIOARC leverages **Neural Architecture Search (NAS)** to explore a **large architecture design space** across various biological data modalities. Evaluate how **architecture**, **tokenization**, and **training strategies** interact and influence performance. BIOARC identifies **high-performing architectures** tailored to biological contexts.

The framework also introduces **architecture prediction methods** that efficiently predict optimal architectures for new biological tasks. The key contributions of the work are:
1. **Principled, automated framework** for designing biological foundation models.
2. **Comprehensive analysis** of the relationship between architecture, tokenization, and training strategies in biological data.
3. **Discovery of novel architectures** that outperform intuition-driven designs.
4. **Set of empirical design principles** to guide future development of biology-specific models.
5. **Architecture prediction techniques** enabling efficient adaptation to new biological tasks.

BIOARC represents a step towards creating **task-specific and general-purpose foundation models** for biology. By moving from intuition-driven to **principled architecture discovery**, it establishes a systematic methodology to advance biological AI research.

**Strengths:**

- The idea of applying techniques in Neural Architecture Search for discovery of patterns and design principles for biological foundation models is novel and interesting
- The paper is very detailed and considers different important aspects of biological foundation models such as tokenizations and architectural design aspects such as type of block used, the width of the network and the depth of the network.
- The paper is very well written and clearly structured in most parts
- The contribution of the paper is very significant and I believe that the insights derived are of general interest for ICLR research community. Furthermore, the architectures derived being much smaller and performant compared to state-of-the-art foundation models, makes these models more accessible and deployable

**Weaknesses:**

- The techniques of weight sharing, random path sampling followed by training of a predictor are quite well studied in the NAS literature and across applications [1,2,3], the application of the techniques for biological foundation models is however novel.
- The experimental setup is not very clear in some parts:
    - Could the author's elaborate on the total compute budget for finetuning architectures? How faster is the convergence of a model initialised from supernet weights v/s a model trained from scratch?
    - Given the large size of the search space (different modules per-layer), could some paths through the network be undertrained?
    - Could the authors plot performance metrics for architectures randomly sampled from the supernet (before finetuning) v/s the parameter count in the architectures?
    - Could the authors compare different search algorithms eg: random-search v/s evolutionary search v/s bayesian-optimization?

**Questions:**

Check weaknesses

---

> ### Author Response · Authors · 2025-11-26
>
> We thank Reviewer KRGZ for the constructive feedback and valuable insights. We are very excited to see that the reviewer finds our contribution to be significant, novel and expressed in detail.
>
> Here we address your comments .
> ***
> ***
> ### **W1 Compute budget**
> We thank the reviewer for pointing out the missing information. We have added a cost analysis section in Appendix A.6.3.***
> ***
> ***
> ### **W2 Potential undertrained paths**
> This is a valid concern inherent to One-Shot NAS methods. However, in BIOARC, we mitigate the risk of under-training through three key design choices and empirically validate the sufficiency of our training via rank correlation analysis:
>
> **Uniform Sampling.** We employ a Single Path One-Shot approach with uniform sampling (Section 3.3) to ensure that every operation within the pruned space has an equal probability of being trained, preventing the collapse towards easier-to-train sub-networks.
>
> **Module-Level Weight Sharing.** Even if specific paths are sampled less frequently, the underlying modules are heavily reused across diverse topologies. A specific module receives gradient updates whenever any path containing it is sampled, ensuring rigorous training at the component level.
>
> **Independent Optimization.** We employ an independent optimization protocol (Section 3.4 ). We evaluate architectures by training from scratch or individual fine-tuning. This decouples our ranking from sampling frequency, ensuring results reflect true topological superiority rather than training artifacts.
>
> **Empirical Validation.** Most importantly, we empirically verified this in Appendix A.6.13. We compared the ranking of architectures derived from the Supernet against the 'ground truth' ranking obtained by training each architecture from scratch. We observed a high Spearman Rank Correlation ($\rho = 0.8170$), as shown in Figure 13. This strong alignment confirms that the Supernet weights are sufficiently trained to accurately capture the relative performance ordering of different candidates.
> ***
> ***
> ### **W3 Performance v/s Parameter**
> We thank the reviewer for this insightful suggestion. To address this, we have performed a new analysis and added the results in Appendix A.6.11 and Figure 12. Specifically, we evaluated the sampled architectures directly using the pretrained supernet weights (before finetuning) as requested. To strictly measure the supernet's representation quality without the interference of task-specific optimization, we froze the backbone weights and only trained a linear classification head.
>
> We plotted the performance of these architectures against their parameter counts in Figure 12. The results reveal two key findings:
>
> **No Positive Correlation.** As shown in the scatter plot, we observe no positive correlation between model size and performance. Larger parameter counts do not guarantee higher accuracy.
>
> **Efficiency over Size.** High accuracy scores are frequently achieved by efficient architectures (indicated by lighter colors in Figure 12) with relatively smaller parameter budgets.
>
> This new experiment strongly validates that the performance gains in BIOARC stem from the superior topological design (inductive bias) of the discovered architectures, rather than simply increased model capacity.
> ***
> ***
> ### **W4 Ablation of search algorithms**
> We appreciate the reviewer's query regarding the choice of search algorithms. We would like to clarify that our choice of a One-Shot Weight-Sharing approach over Multi-Run iterative methods (e.g., Evolutionary Search, Bayesian Optimization) is driven by the fundamental constraints of pretraining biological foundation models.
>
> **The Necessity of One-Shot for Pretraining**
>
> Neural architecture search methods fundamentally fall into two categories. ***Multi-Run Iterative Search Algorithms*** like Evolutionary Search or Bayesian Optimization typically require training and evaluating thousands of candidate architectures to guide the search. One-Shot Evaluation requires training on a single Supernet where all candidates share weights. For our goal of pretraining, the computational cost of "Multi-Run" methods is prohibitive. Pretraining a single model on large-scale genomic data (e.g., full human genome) takes hundreds of GPU hours. Repeating this for every candidate in an evolutionary loop is computationally infeasible. Therefore, the One-Shot Supernet approach is the only viable path to efficiently explore the space without incurring the cost of repetitive pretraining.
>
> **Architecture prediction methods**
> The advantage of both methods mentioned are able to provide an accurate prediction of model performance, which could be achieved by our architecture prediction methods.
>
> **Future Work**
> While we focused on the One-Shot paradigm to make pretraining feasible, we agree that exploring how Bayesian Optimization or other multi-run methods could assist in the pruning phase of the Supernet itself is a promising avenue for future work.
> ***

---

### Official Review · Reviewer_eZuN · 2025-11-08

**Soundness:** 2
**Presentation:** 1
**Contribution:** 3
**Rating:** 2
**Confidence:** 3

**Summary:**

This paper introduces BIOARC, a Neural Architecture Search (NAS) framework designed to discover optimal neural architectures for biological foundation models. The authors motivate their method by observing that current biological models directly repurpose NLP architectures like Transformers without accounting for the unique properties of biological sequences -- sparse motifs, long-range dependencies, and physicochemical constraints that differ from natural language. BIOARC addresses this by first defining a search space spanning five module types (CNN, LSTM, Transformer, Mamba, Hyena) with various depths and hidden dimensions, yielding 67 million candidate architectures. This space is aggressively pruned to 360 paths using heuristic rules. Then, a weight-sharing supernet is constructed from the remaining unique layers and trained via one-shot uniform path sampling on large-scale biological corpora (DNA and proteins). They consider two self-supervised objectives: masked modeling and contrastive learning.

Additionally, to predict architectures for new tasks, the paper proposes three methods: a Graph Neural Network predictor that encodes architectures as graphs and tasks via pretrained language model embeddings; a direct LLM prompting approach; and a multi-agent BIOARC Agent system that chains specialized LLMs that are assigned specific roles to reason over task similarity and performance data. However, details on this part are highly unclear and I am not sure what the point of this section.

On 12 DNA tasks and 6 protein tasks, BioArc shows good performance. The authors attempt to answer six core research questions about competitiveness, architectural patterns, scaling laws, training strategies, tokenization effects, and prediction methods. The key results are that BioArc-discovered architectures (3-5M parameters) achieve 8-15 point absolute gains over two larger baselines (DNABERT-2 and Neucliotide Transformer) across all GUE tasks, demonstrating that a well-chosen architecture can compensate for scale. The paper identifies a consistent pattern where top DNA architectures cascade through Hyena -> Transformer -> CNN modules, and reveals architecture-tokenization patterns. However, on protein benchmarks, these architectures are strong on sequence tasks but fail on structure prediction (15.88% vs. ESM-1b's 28.17%).

Overall, I found it quite difficult to parse the paper and results, as it is not well-presented and often incomplete in both the main text and the appendix.

**Strengths:**

In my reading, I found the presented results to be impressive. The authors also uncovered useful and interesting architectural insights.

1. On the DNA-based GUE benchmark, BIOARC achieves **8-15 point absolute improvements** over established baselines (DNABERT-2, VQDNA, Nucleotide Transformer) across 12 diverse genomic tasks. These gains represent a clear improvement in performance on a standardized benchmark, achieved with models that are **much** smaller in parameter count and trained on substantially less pretraining data.

2. The paper identifies an actionable design principle -- optimal tokenization is architecture-dependent in biological sequences. Transformers perform best with 6-mer tokenization, while CNNs peak with 1-mer tokenization. Interestingly,  this interaction reverses under pretraining. This is empirically well-validated across multiple tasks and provides immediate practical guidance for practitioners building biological models.

3. The supernet approach (after intelligent pruning) is a smart idea borrowed from the original paper. It successfully identifies high-performing mixed architectures (e.g., Hyena -> Transformer -> CNN) that outperform single-module designs. This pattern can be easily adopted by others who are training their own biological sequence model.

4. Finally, it is nice to see that similar downstream tasks benefit from architectures that are similar to each other (Figure 7).

Overall, I see this paper as a useful investigation into the relationships between architectures and downstream tasks in biological sequence modeling.

**Weaknesses:**

1. The biggest weaknesses of this paper are the presentation, writing, and clarity. It was quite hard for me to parse the method in its entirety, as well as the presentation of the various training configurations of the BioArc models (only-ft, ...). In fact, the paper has several places where the writing is extremely concise, to the point of being a blurb (eg, A.8.6, L1187). A prime example of writing sloppiness can be seen in Appendix A.8.7 (L1271), where the section is woefully incomplete and stops mid-sentence, and A.5 L943, where it is unclear what the sentence says.

2. The BioArc methodology presentation keeps jumping around the **exact way** to rank the candidates. The initial pruning methodology is clear (I expect an explicit pruning algorithm, complete with hyperparameters, to be released in the codebase upon publication/deanonymization). However, following this initial pruning, how exactly the candidates are ranked is not clear to me. The paper implies that all 360 pruned architectures are evaluated via fine-tuning or training from scratch on each downstream task, but the precise algorithm for scoring, ranking, and selecting the "best" architectures is never explicitly detailed. The supernet is never shown to be used for ranking. This information seems to be missing entirely, and for a NAS method, this component is critical. The paper never defines the ranking objective: is it validation accuracy, test accuracy, or a Pareto frontier? Without this, the 'top-5 architectures' in Figure 2 and the selection for Figure 3 are **arbitrary and irreproducible**.

3. Following the above point, the architecture prediction section and contribution seems to be conecptually orthogonal to the rest of the methodology. It does not seem to be a core component, and is not tied into the methodology as well as the other components. Furthermore, the results in Table 3 are missing the evaluation context in A.8.7 entirely, and therefore, I cannot consider these results grounded.

4. Unconvincing (and sometimes missing) diagnosis of results and their interpretation -- in Sec. 4.2, L318 is an entirely generic sentence, and L323 is not a (contextually) convincing reason for the underperformance of the BioArc models, when compared to ESM-1b.  Furthermore, ESM-2 is overlooked as a baseline, despite being released in 2023.

5. (Minor but still important) The figures presented in the paper are of low quality; for example, see Figs. 4,10. There are also some typos littered throughout the paper (L316, L295).

Considering the unclear writing, confusing presentation, and underexplained sections, I recommend **rejection** at this stage of the submission. This initial decision is despite the paper's meaningful results. I suggest that the authors either **substantially** improve writing and presentation in their rebuttal (especially making the methodology clearer and presenting the missing parts) or consider resubmission after substantial rewriting, in which case the results and insights can really shine with a clearer presentation of the methodology.

**Questions:**

1. I find the choice of masked modeling and contrastive learning alone as training tasks interesting. Several biological sequence models (EVO 1/2 being a salient example) are now being trained using next-token-prediction objective. Is there a reason why this was not considered?

2. Why were the EVO models not discussed at all in the paper?

3. Why was the ESM-2 paper not considered as a baseline?

3. After the supernet is trained and yields 360 pretrained paths, are all of these architectures trained from scratch for the final evaluations? This is unclear, both in the main text and appendix. In case I missed this part, please point me to the relevant sections of the text.

4. Is it reasonable to assume that rankings from supernet paths are muddled with gradient sharing between connected other paths and training objective? Did you perform a rank correlation analysis between performance of the supernet paths and the same paths trained from scratch?

---

> ### Author Response · Authors · 2025-11-26
> **Part 1**
>
> We thank Reviewer eZuN for the constructive feedback and valuable insights. We are encouraged that the reviewer finds our work to be a useful investigation into biological sequence modeling with actionable design principles. We acknowledge the concerns regarding the writing clarity and have carefully polished the presentation to ensure our ideas are conveyed more effectively.
>
> Here we provide detailed response to each points:
> ***
> ***
> ### **W1/W2. Concerns on Presentation, Writing, and Clarity on methodology**
> We sincerely apologize for the difficulty and frustration caused by the presentation issues in our initial submission. We have taken your feedback very seriously and have conducted a thorough revision of the methodology to improve clarity and correct errors. We have significantly revised the original Evaluation Protocol paragraph into a whole new section (Section 3.4) and added another new section Evaluation Setting Details in Appendix A.6.2.
>
> **Method clarification.** To address the question of "how exactly candidates are ranked", we have added a new paragraph Architecture Ranking Strategy in Section 3.4. Our ranking strategy leverages test set performance. Since our benchmark includes tasks with distinct metrics (e.g., Accuracy vs. RMSE) and varying statistical distributions,  we use Z-score normalization to correct differences in metric scales and optimization directions among the tasks, detailed in Equation (8).
>
> **Training configuration.** The revised 'Evaluation Protocol' paragraph in Section 3.4 now explicitly details the BioArc training configurations. Specifically, we employ two distinct settings for evaluating each path in the supernet: (a) initializing from pretrained supernet weights followed by fine-tuning (Equation 6), and (b) random initialization followed by training from scratch (Equation 7)，ensuring a fair assessment of each architecture's topological potential. Additionally, we have included a new section in the Appendix, 'Evaluation Setting Details,' to provide clearer notation.
>
> **Missing context.** We made the following refinement in the paper.
> 1. We significantly refined Section More Results on Architecture Prediction Method (previous Appendix A.8.7), turning it into Section Architecture Prediction Experiment Settings and More Results (Appendix A.6.12). In this section, we have now demonstrated clear definitions, settings and complete analysis.
>
> 2. We also strengthen the analysis in Section More Results on Different Tokenizers(Appendix A.6.9) to avoid being blurb. This section now describes the ablation studies on different LLM backbones and provides the critical experimental setup for the results in Table 4
> and Table 11.
>
> 3. The example in Section Architecture Encoding Example (Appendix A.5) is also refined.
> ***
> ***
> ### **W3. Contribution of Architecture Prediction**
> We thank the reviewer for highlighting the structural disconnect and the missing appendix content. We address these two points below:
>
> **Restoration of Missing Context in Appendix A.8.7.**
> We sincerely apologize for the oversight that caused the text in Appendix A.8.7 to be truncated in the original submission.  We have fully restored the complete analysis and experimental context in the revised manuscript. We assure the reviewer that the results in Table 3 are grounded in the fair settings now detailed in the corrected Appendix A.6.12.
>
> **The Integral Role of Architecture Prediction**
> We acknowledge that our original exposition failed to explicitly articulate the logical dependency between the search and prediction components. Rather than functioning as an independent add-on, the Architecture Prediction module is directly grounded in our core empirical observation that optimal architectures exhibit significant topological similarity across functionally similar tasks (Section 4.2). Driven by this insight, we designed the prediction module as a functional bridge to translate our rigorous search into a scalable utility. The connection relies on the following logic:
> 1. Repeating the search is not strictly necessary: While evaluating 360 pruned architectures is computationally feasible, mandating this exhaustive process for every new task introduces avoidable overhead when the optimal design can be predicted directly.
> 2. We leverage the search as a one-time investment: We position the exhaustive NAS process as a foundational step to construct a ground-truth knowledge base, allowing the BioArc Agent to act as an instant inference engine.
> 3. This enables efficient generalization: This design enables the system to generalize findings to unseen tasks without incurring the computational debt of re-evaluation. To rigorously validate this, we established both Supervised and Transfer settings, explicitly comparing the predictor's performance when similar tasks are present in the existing knowledge base versus when generalizing to novel domains.
> ***
> ***

---

> > ### Author Response · Authors · 2025-11-26
> > **Part 2**
> >
> > Part 2
> > ***
> > ***
> > ### **W4\Q3. Unconvincing diagnosis of results and their interpretation\Missing baseline**
> > **Inclusion of ESM-2 Baselines.**
> > We have updated Table 2 to include both ESM-2 (650M) and ESM-2 (8M). The results reinforce our findings: BioArc ($\sim$3M) remains highly competitive on physicochemical tasks (Solubility, PPI) against ESM-2 (650M) but faces a performance gap on structural tasks.
> >
> > **Deepened Analysis.**
> > We have completely rewritten the analysis in Section 4.2 for better interpretation on our results.
> >
> > ***Why BioArc wins on Sequence-Function tasks.*** We attribute this to Task-Specific Inductive Bias. Our NAS-discovered hybrid topologies physically align better with local physicochemical signal densities than pure architectures, enabling high data efficiency.
> >
> > ***Why BioArc trails on Structure tasks.***
> > We explicitly identify two limiting factors:
> >
> > *Scale of Pretraining Data*: Structure prediction relies on memorizing the global evolutionary landscape for remote homology detection. Unlike foundation models trained on the full UniRef/BFD databases, \our utilizes a subset for search efficiency, limiting this global memorization.
> >
> > *Spatial Inductive Bias*: We highlight that the small ESM-2 8M outperforms the much larger ProtBERT on structural tasks, even though ProtBERT is trained on a bigger dataset. This effectively isolates the impact of Rotary Positional Embeddings used in ESM-2, which explicitly model relative distances. In contrast, ProtBERT and our current search space rely on absolute positioning. Lacking this explicit spatial modeling creates a performance ceiling for structural modeling, regardless of parameter scale.
> >
> > We hope this revised diagnosis offers a grounded, technical, and convincing interpretation of the results.
> > ***
> > ***
> > ### **W5. Figure quality and typo**
> > We thank the reviewer for their attention to detail. We have addressed these presentation issues in the revised manuscript.
> > For figure quality, we have added the numeric values on the right panel of Figure 4 and Figure 10 to ensure clarity and readability. We also have replaced Figure 7 with a high resolution version. We have corrected the specific typos pointed out (e.g., L295, L316) to improve readability.
> > ***
> > ***
> > ### **Q1.Choices of pretraining strategy**
> > Thank you for this valuable feedback. We agree that the success of recent generative models (e.g., Evo 1/2) highlights the importance of Next Token Prediction (NTP).
> >
> > **Rationale for Initial Selection**
> > Our initial focus on MM and CL was driven by the alignment between training target and task. Our downstream benchmarks are primarily discriminative (e.g., promoter detection, protein solubility). For such tasks, bidirectional representations (like BERT/ESM) and contrastive learning typically offer a stronger inductive bias than autoregressive ones (like GPT/Evo).
> >
> > **New Experiments with NTP**
> > Motivated by your suggestion, we have conducted additional experiments using an NTP objective. These results have been included in the revised manuscript in Table 1 (and Section 3.3). We observed that while NTP is effective, it generally performs slightly below MM and CL on our specific benchmarks. This aligns with the intuition that for discriminative tasks (e.g., classification/regression), the bidirectional context utilized by MM and the global sequence representations optimized by CL is often more sample-efficient than the unidirectional context of NTP.
> > ***
> > ***
> > ### **Q2.Missing discussion of EVO models**
> > We have added a discussion on EVO models in the revised manuscript. While our primary focus was on sequence embedding, which aligns with our selected tasks and baselines, we agree that discussing EVO models provides valuable context and have updated the related work section accordingly.
> > ***
> > ***
> > ### **Q3.ESM-2 as additional baseline**
> > We have added ESM-2 as a baseline with detailed analysis in Section 4.2.
> > ***
> > ***
> > ### **Q4. Experiment Settings**
> > Originally it was in a paragraph Evaluation Protocol in Section 3.3. We have significantly revised the paragraph into a whole new section (Section 3.4) and added another new section Evaluation Setting Details in Appendix A.6.2.
> > ***
> > ***
> > ### **Q5. Rank correlation analysis**
> > We have added the analysis on Appendix A.6.13. From Figure 13, we observe a strong positive correlation, with data points tightly clustered around the diagonal line ($y=x$), which represents perfect ranking agreement. This high correlation coefficient ($\rho=0.8170$) demonstrates that despite the parameter sharing inherent in the supernet, the BIOARC supernet effectively captures the relative performance ordering of different topologies without significant interference.

---

### Official Review · Reviewer_r4ru · 2025-11-09

**Soundness:** 3
**Presentation:** 4
**Contribution:** 3
**Rating:** 6
**Confidence:** 4

**Summary:**

This paper introduces BIOARC, a novel framework designed to automate the discovery of optimal neural network architectures for biological foundation models (FMs). The authors argue that the common practice of directly repurposing architectures from general AI domains (e.g. transformers in NLP and Vision) is suboptimal for biological data, which has unique properties such as long-range dependencies and complex "grammars" dictated by physicochemical laws. BIOARC leverages Neural Architecture Search (NAS) to systematically explore a vast and diverse design space that includes modern modules like CNNs, Transformers, Mambas, Hyenas, and LSTMs across multiple biological data modalities (DNA and proteins).

The paper's core contributions are:
* A principled methodology for discovering high-performing, task-specific architectures for biological data through NAS using modern modules. The search is made computationally feasible by leveraging a weight-sharing supernet.
* A systematic analysis of the complex interplay between neural architecture, tokenization strategy, and training strategy (e.g., pretraining vs. training from scratch)
* Empirical results demonstrating that the discovered architectures are often significantly smaller (e.g. ~25x) yet outperform larger, established foundation models on a variety of DNA and protein tasks.
* The development of a "BIOARC Agent," an LLM-based system to predict optimal architectures for new biological tasks.

**Strengths:**

* **Problem Significance**: The paper correctly identifies a key bottleneck in computational biology: the reliance on intuition-driven or repurposed architectures. The move towards a principled, automated discovery process is a strong and important research direction.
* **Comprehensive Analysis**: The study's greatest strength is its systematic evaluation. It does not just search for a single best model but rigorously analyzes the "interplay between architecture, tokenization, and training strategies". This yields durable insights, for example:
* * The optimal tokenizer is highly architecture-dependent (e.g., in Figure 5, CNNs perform best with 1-mer while Transformers prefer 6-mer).
* * Pretraining is not universally beneficial; training from scratch (only-ft) achieves the highest "win rate" for several tasks (Figure 4).
* **Strong Empirical Results**: The BIOARC-discovered architectures are highly competitive. They achieve state-of-the-art performance on all 12 DNA tasks while being significantly smaller (~4.5M params) than baselines like DNABERT-2 (117M params). This demonstrates the effectiveness of the search methodology.
* **Novel Architectural Insights**: The analysis reveals that the best-performing models are hybrid architectures, such as the Hyena -> Transformer -> CNN pattern identified for DNA tasks (Figure 2). This is a valuable finding, suggesting that combining modules with different inductive biases is crucial in the biological domain.

**Weaknesses:**

My main concerns are focused on the clarity of the supernet training methodology and the presentation of some results.
* **Major Question on Supernet Training Dynamics**: The paper adopts a "Single Path One-Shot" approach where a single path is sampled and updated in each step. This raises two critical questions about training stability and fairness:
* * Training Imbalance: In a vast search space, some shared blocks (e.g., a final-layer block) may be part of significantly more candidate paths than other blocks (e.g., a unique first-layer block). How do the authors ensure balanced and fair training, such that commonly shared blocks are not disproportionately over-trained compared to less common ones?
* * Incoherent Updates: A shared block's weights are updated based on an input representation from a stochastically sampled previous block. This means the block is not trained on a stable input distribution but on a "moving target" from a mix of different potential predecessors. How does the supernet converge reliably under this training scheme? A deeper discussion beyond "decouples the training"  is needed.
* **Ablation of Discovered Architectures (Fig. 2)**: The top DNA architectures shown in Figure 2 start with a Hyena block, followed by Transformer and CNN blocks. Given Hyena's strength in capturing long-range dependencies, how do we know the subsequent blocks are contributing significantly? Is it possible that the Hyena block is doing most of the work?
* **Impact of Network Depth (Fig. 2)**: The top 5 DNA architectures in Figure 2 all appear to have a depth of 6. Does this imply that the deepest allowed model was always optimal? Some discussion on the impact of depth on performance would be valuable.
* **Clarity of Research Questions**: RQ3 ("Does a scaling up of our optimal architecture yield a better foundation model outperform existing...") seems to overlap significantly with RQ1 ("How competitive are the BIOARC discovered architectures against established, large-scale foundation models?"). While the distinction appears to be about small vs. scaled-up discovered models, this could be clarified.

**Questions:**

Other than the questions mentioned above, I would also like to give some suggestions related to the clarity rather than the flaws in the overall manuscript.
* **Reporting Training cost**: The paper demonstrates a powerful methodology, but it would be helpful to quantify the computational cost. How long did the supernet pretraining take, and on what hardware? This context is important for assessing the framework's practicality.
* **Figure Clarity**:
* * Figure 4 (Right Panel): This stacked bar chart is very difficult to interpret. The bars are of similar lengths, and the segments are hard to compare visually. A simple table reporting the percentage "win rate" for each of the architectures for each task would be much clearer.
* * Figure 5: The x-axis is unlabeled. I infer it represents the Task Index (0-11) from Appendix A.1.1. Using a line plot is confusing, as it implies a sequential or time-series relationship between the tasks, which does not exist. A grouped bar chart or dot plot would be a more appropriate visualization.
* * Most figures: The legend and axis labels in many figures are too small and require significant zooming to be legible.
* **Typos/Phrasing**:
* * Line 373: "To overcome this challenge, To overcome this," is repetitive.
* * Line 375: "...input dimension of and a 512..." – the "of" appears to be a typo.
* * Line 546: The grammar of RQ3 ("...a better foundation model outperform exsiting...") should be revised for clarity.
* * Line 948: "at at BioArc-9794" is a repetition.

---

> ### Author Response · Authors · 2025-11-26
> **Part 1**
>
> We sincerely appreciate the reviewer's acknowledgement of the significance of the problem we address. We are also grateful for the recognition of our comprehensive analysis regarding architecture, training strategy, and tokenization. Furthermore, it is highly encouraging that the reviewer recognized the effectiveness of our framework and the insights it provides.
>
> Here we provide a detailed response to each point.
> ***
> ***
> ### **W1. Major Question on Supernet Training Dynamics**
>
> **Training Imbalance**
>
> We appreciate the reviewer's observation regarding the potential training imbalance caused by our monotonic width constraint. This constraint naturally leads to fewer valid candidates in deeper layers (e.g., Layer 1 might have 30 choices, while Layer 6 might have 10). Since we adopt uniform sampling at the path level over the pruned search space, modules that appear in more paths are indeed trained more frequently. We address this via two key designs:
>
> ***Alignment with Module Size.*** We suggest this distribution is beneficial. Deeper layers are constrained to be wider and contain significantly more parameters. Thus, their higher sampling frequency naturally aligns with the greater training volume required for their convergence.
>
> ***Independent Optimization.*** Crucially, to eliminate supernet bias, we employ an independent optimization protocol (Section 3.4 ). We evaluate architectures by training from scratch or individual fine-tuning. This decouples our ranking from sampling frequency, ensuring results reflect true topological superiority rather than training artifacts. We conduct analysis on the ranking correlation of ranking consistency between initialized from supernet weight and the performance obtained by training from scratch. We observed high correlation coefficient, which means the supernet training is appropriate.
>
> **Incoherent Updates**
>
> The "incoherent update" you observed is precisely the feature of the Single Path One-Shot method, specifically intended to solve this "co-adaptation" problem. As analyzed by [Bender et al](https://proceedings.mlr.press/v80/bender18a/bender18a.pdf), "co-adapt" training leads to a catastrophic failure: the weights of a shared block (e.g., $Block_A$) become over-specialized to its most common upstream block (e.g., $Block_B$). Our method intentionally introduces this "incoherence" via "uniform path sampling". As stated in the foundational paper by [Guo et al](https://arxiv.org/pdf/1904.00420), this method alleviates the weight co-adaption problem. Therefore, this "incoherent update" serves not as an obstacle to convergence but rather as the key regularization technique required to prevent "co-adaptation."
> ***
> ***
> ### **W2. Ablation of Discovered Architectures**
> We appreciate the reviewer’s hypothesis. Given Hyena’s strong performance on long sequences, it is logical to question if it dominates the inference process while the Transformer and CNN modules remain passive. However, we have conducted two specific analyses to confirm that the effectiveness of BioArc stems from the synergy of mixed modules, rather than Hyena alone.
>
> **Comparison with pure architecture.**
> To fairly evaluate the strength of the hybrid architecture, we replicated our architecture search but restricted the search space to single module types. This allowed us to identify the optimal 'Pure' baselines (e.g., Pure Hyena, Pure Transformer) for a fair comparison. As shown in Figure 8, BioArc outperforms these optimized pure architectures, **including Hyena**. We also compared with pretrained **HyenaDNA** in Table 2, in which BioArc is consistently better. These observations confirm that the performance gains stem from the specific combination of modules. Details could be found in Appendix A.6.6.
>
> **Test on each layer.**
> To verify the contribution of each module, we trained separate classification heads on the output of each layer in our top-performing architecture to evaluate intermediate feature quality . The observed monotonic improvement in accuracy confirms that the diverse modules collaboratively refine representations, validating the effectiveness of the holistic hybrid design. Details could be found in Appendix A.6.7.
> | | Layer 0 (Hyena) | Layer 1(Transf) | Layer 2(Transf)| Layer 3 (CNN) | Layer 4 (CNN) | Layer 5 (CNN) |
> | :--- | :---: | :---: | :---: | :---: | :---: | :---: |
> | Accuracy (%) | 70.32 ± 0.89 | 74.07 ± 0.86 | 75.09 ± 0.82 | 78.36 ± 0.62 | 82.09 ± 0.49 | 83.18 ± 0.46 |
>
> **Interpretation Analysis.**
> We have conducted interpretation analysis on each layer and found that each layer have different role to capture the inductive bias. Detailed process and results could be found in Appendix A.7.
>
> ***
> ***

---

> > ### Author Response · Authors · 2025-11-26
> > **Part 2**
> >
> > ***
> > ***
> > ### **W3. Impact of Network Depth**
> > We appreciate the reviewer’s observation regarding the prevalence of depth-6 models among the top candidates. To thoroughly address whether "deeper is always optimal," we performed complementary analyses as detailed in Appendix A.6.10
> >
> > **Performance of different depth.** We conduct statistical analysis on accuracy of 3-6 layers architectures across all 12 DNA downstream tasks as shown in Table 9. While the median performance of 6 layer architectures is highest, 4 layers have the highest mean accuracy. We also observe in Figure 6 that optimal architectures for Protein are not always 6 layers. This implies that depth is not the sole determinant of performance but rather the structural composition of the architecture is key.
> >
> > **Structural consistency across depths.** We observe in Figure 11 that the top-performing 4- and 5-layer architectures exhibit structural patterns similar to those of the 6-layer models. This indicates that once an effective architectural pattern is identified, scaling up the depth can further enhance performance.
> >
> > | Depth | Count | Mean (%) | Std | Min (%) | Max (%) | Median (%) |
> > | :--- | :---: | :---: | :---: | :---: | :---: | :---: |
> > | 3 Layers | 60 | 81.00 | 4.23 | **53.77** | 84.85 | 81.49 |
> > | 4 Layers | 100 | **81.74** | 5.57 | 53.38 | 85.14 | 83.07 |
> > | 5 Layers | 100 | 77.74 | 10.24 | 47.44 | 85.48 | 82.56 |
> > | 6 Layers | 100 | 78.17 | 10.72 | 47.70 | **86.53** | **83.30** |
> >
> > **Scalability of the optimal architecture.** We expanded discovered optimal architecture into a 74.7M-parameter foundation model by stacking 2 Hyena, 4 Transformer, and 3 CNN layers with a hidden dimension of 1024. The bigger FM yields improvement on 10/12 tasks compared to smaller versions as shown in the following Table. This demonstrates that our discovered architecture effectively supports scaling.
> >
> > | Model | 0 | 1 | 2 | 3 | 4 | 5 | 6 | 7 | 8 | 9 | 10 | 11 | Avg. |
> > | :--- | :---: | :---: | :---: | :---: | :---: | :---: | :---: | :---: | :---: | :---: | :---: | :---: | :---: |
> > | VQDNA (HRQ) | 72.48 | 76.43 | 66.85 | 58.92 | 78.10 | 71.02 | 70.58 | 78.50 | 90.75 | 94.48 | 74.52 | 89.53 | 76.85 |
> > | BioArc-FM (4.79M) | 82.50 | 85.10 | 76.40 | 72.40 | 82.20 | 80.47 | 83.34 | 76.35 | 90.24 | 95.25 | 79.93 | 68.26 | 81.04 |
> > | BioArc-FM (74.7M) | 84.30 | 83.60 | 85.70 | 78.80 | 88.60 | 83.09 | 82.70 | 86.62 | 92.28 | 96.01 | 79.61 | 84.83 | 85.51 |
> > | BioArc (task-spec) | 84.70 | 85.90 | 86.20 | 77.70 | 90.10 | 83.87 | 84.51 | 90.05 | 94.38 | 96.46 | 84.34 | 91.17 | 87.45 |
> > ***
> > ***
> > ### **W4 Clarity of Research Questions**
> > We thank the reviewer for pointing out the ambiguity. We have revised the research questions as follows:
> >
> > ***RQ1 (Task-specific Validation)***
> > How competitive are the BioArc architectures against large-scale foundation models when initialized from the pretrained supernet and finetuned? Moreover, how do they perform when trained from scratch?"
> >
> > ***Clarification:*** This question focuses on validating the effectiveness of search architectures with NAS pretraining/training from scratch in a task-specific style.
> >
> > ***RQ3 (Foundation Model Scaling)***
> > Does large-scale pretraining on our optimal architecture yield a foundation model that outperforms existing biological foundation models?
> >
> > ***Clarification:*** This question focuses on exploring the potential of the identified optimal architecture as a foundation model.
> > ***
> > ***
> > ### **Q1. Training cost**
> > We thank the reviewer for pointing out the missing information. We have added a cost analysis section in Appendix A.6.3.
> > ***
> > ***
> > ### **Q2&3 Figure Clarity & Typos/Phrasing**
> > We thank the reviewer for the valuable feedback. We have added precise values in the Figure 4 right panel and labeled the x-axis Figure 5. We have also fixed these typos accordingly.
> > ***

---

### Author Response · Authors · 2025-11-26
**Thanks to all reviewers**

We sincerely thank all reviewers for their time and dedicated efforts in reviewing our work. We are deeply grateful for the constructive and insightful feedback received through this conference, which has significantly helped us refine and strengthen the quality of BioArc.

We are encouraged that the reviewers recognized the value of our work. Specifically:

* **Significance & Novelty**: Reviewers acknowledged that moving from intuition-driven design to principled, automated discovery is a "strong and important research direction" and that applying NAS to biological foundation models is "novel and interesting" and is “a smart idea”. (Reviewer r4ru, KRGZ, eZuN)
* **Strong Empirical Results**: Reviewers commended our models for achieving state-of-the-art performance while being "significantly smaller" and "more accessible" than established baselines like DNABERT-2 and ESM-1b (Reviewers r4ru, eZuN, Rr5N, KRGZ).
* **Systematic Analysis**: The rigorous evaluation of the interplay between architecture, tokenization, and training strategies was highlighted as a "greatest strength" that provides "durable insights" (Reviewer r4ru, KRGZ, Rr5N).
* **Actionable Design Principles**: Reviewers specifically valued our finding that hybrid architectures (e.g., Hyena->Transformer->CNN) could outperform single module designs and could be easily adopted for the community. (Reviewers eZuN, KRGZ).
* **Deep Insights**: Reviewers recognize that we provide interesting insights including the dependence of tokenizer choice on architecture and architecture and similar architectures for similar tasks. (Reviewer Rr5N, eZuN)

In response to the reviewers' constructive comments, we have extensively revised our manuscript. The major updates in the new version include:

**Related Work**
* We added a paragraph to discuss existing hybrid architectures
* We added discussion of EVO Model in Foundation Models for Biological Data paragraph

**Methodology**
* We added Next Token Prediction as a third self-supervised learning objective to the supernet pretraining section and provided its mathematical formulation in Appendix A.4.
* We expanded our evaluation protocol into a new Section 3.4 with clearer descriptions. We added explicit equations for the scoring and ranking strategy (Eq. 8).
* We clarified our motivation for the architecture prediction method in Section 3.5.
* We provided a clear process of how to use LLM for architecture prediction.
* We added a new Table 1 in Section 3.5 to explicitly define the inputs, outputs, and workflows for each role in the BioArc Agent system.
* We added one paragraph Architecture Prediction Evaluation in Section 3.5 to explain the experiment design.

**Experiments**
* We added ESM-2 (1b & 8m) as strong baselines for protein tasks in Table 3.
* We added Next Token Prediction results to Table 2.
* We significantly expanded the interpretation of our findings in the Main Results section. (L374-377 414-422)

**Appendix**
* **Computational Cost**: We added Section A.6.3 and Table 7 to report detailed costs of supernet pretraining, foundation model pretraining and task-specific finetuning.
* **Layer-wise Contribution Analysis**: We added Section A.6.7 and Table 8 to analyze the specific contribution of each layer to the overall performance.
* **Architecture Depth Analysis**: We added Section A.6.10 and Table 9 to investigate the statistics and impact of different network depths on model performance.
* **Parameter Scaling**: We added Section A.6.11 and Figure 12 to analyze the relationship between model size and accuracy, proving that performance gains stem from architectural topology rather than just parameter count.
* **Interpretation Analysis**: We conducted a case study to validate if BioArc discovered architecture capture underlying biological grammar on a certain task.

**Overall Writing and Figures**
* We made significant revisions on our methodology presentation and extended Appendix for more and clearer analysis.
* We proofread the manuscript and fixed typos highlighted by the reviewers, including repetitive phrases and grammatical errors.
* We restored the previously incomplete text in the Appendix regarding Architecture Prediction (now Appendix A.6.12), providing a full discussion of the ablation study and results.
* We increased the font size of axis labels and legends in key figures (e.g., Figure 3 4 5 9 10). We replaced Figure 7 with a high resolution version. We included precise numerical values in the right panels of Figure 4 and Figure 9 for better readability.

In the following threads, we provide detailed point-by-point responses to address the specific "Weaknesses" and "Questions" raised by each reviewer.

---

### Meta-Review · Area_Chair_3PJY · 2026-01-06

**Summary:**

* Missing technical details: The most important criticisms by reviewers centered on how the Neural Architecture Search (NAS) truthfully functioned and how winners were selected; Specifically, multiple reviewers stated that the paper did not explicitly define how the *top* architectures were ranked. It was not clear if the ranking was based on validation accuracy, a Pareto frontier (balancing size vs. performance), or another metric.

* Baseline selection: While the paper claimed SOTA results, multiple reviewers identified several gaps in the empirical evaluation. Example is ESM-2, a standard and more powerful version of the ESM protein model, in favor of the older ESM-1b

* Lack of scientific discussion: Some Reviewers also raised the point that the paper lacked a deep biological explanation for why a specific hybrid (Hyena -> Transformer -> CNN) was superior, which is critical for a "Bio-centric" paper!

* Generalizability of principles: While the authors claim "general design principles," these are derived from a specific set of benchmarks. Reviewers remain skeptical about whether these principles would hold true for more niche biological tasks, such as RNA-folding or small-molecule interaction, which were not part of the search space.

**Reviewer Concerns:**

During the rebuttal period, the authors revised the manuscript to clarify some of the issues raised and concerns. The authors successfully moved the paper from "technically incomplete" to "technically sound." However, they struggled to bridge the gap from "a good NAS paper" to "a transformative biological discovery."

**Reviewer Scores:**

* The reviewer Rr5N gave 2 as a score with confidence of 2

* The reviewer KRGZ suggested a score of 6 with confidence of 3

* The reviewer ZuN rated 2  with confidence of  3

* The reviewer r4ru suggested a rating of 6 with confidence of 4.


While all reviewers were actively engaged during the rebuttal phase and had a chance to increase their scores or at least mention this, they remained ultimately unconvinced that the authors had fully addressed the fundamental concerns regarding baseline comparisons and biological interpretability.

---

### Decision · Program_Chairs · 2026-01-26

Reject